# Fast Convergence of $\Phi$-Divergence Along the Unadjusted Langevin Algorithm and Proximal Sampler

**Siddharth Mitra**                                                             SIDDHARTH.MITRA@YALE.EDU
**Andre Wibisono**                                                             ANDRE.WIBISONO@YALE.EDU
*Department of Computer Science, Yale University, New Haven, CT, USA*

**Editors:** Gautam Kamath and Po-Ling Loh

## Abstract

We study the mixing time of two popular discrete-time Markov chains in continuous space, the Unadjusted Langevin Algorithm and the Proximal Sampler, which are discretizations of the Langevin dynamics. We extend mixing time analyses for these Markov chains to hold in $\Phi$-divergence. We show that any $\Phi$-divergence arising from a twice-differentiable strictly convex function $\Phi$ converges to $0$ exponentially fast along these Markov chains, under the assumption that their stationary distributions satisfy the corresponding $\Phi$-Sobolev inequality, which holds for example when the target distribution of the Langevin dynamics is strongly log-concave. Our setting includes as special cases popular mixing time regimes, namely the mixing in chi-squared divergence under a Poincaré inequality, and the mixing in relative entropy under a log-Sobolev inequality. Our results follow by viewing the sampling algorithms as noisy channels and bounding the contraction coefficients arising in the appropriate strong data processing inequalities.

**Keywords:** Langevin dynamics, unadjusted Langevin algorithm, proximal sampler, $\Phi$-divergence, $\Phi$-Sobolev inequalities, strong data processing inequalities

## 1. Introduction

Sampling from a probability distribution is a fundamental task that appears in many fields, including machine learning, statistics, and Bayesian inference (Gelman et al., 1995; MacKay, 2003; Robert et al., 1999; Von Toussaint, 2011; Johannes and Polson, 2003). Suppose we wish to obtain samples from a continuous probability distribution $\nu \propto \exp(-f)$ on $\mathbb{R}^d$; a common approach is to construct a Markov chain which admits $\nu$ as its invariant or stationary distribution, and then draw samples after some initial burn-in time. A rigorous study of burn-in time (Gilks et al., 1995; Geyer, 2011) leads one to analyze the *mixing time* of a Markov chain, which tracks how quickly the Markov chain converges to its stationary distribution (Levin et al., 2017). Different choices of distances[1] between probability distributions lead to different guarantees, which can be bounded in terms of each other, to yield interesting results and bounds.

A general, useful, and well-studied family of divergences are that of $\Phi$-divergences (Csiszár, 1967)[2], which include many popular divergences such as Kullback-Leibler (KL) divergence, chi-squared divergence, Total Variation (TV) distance, and squared-Hellinger distance. For any convex function $\Phi : \mathbb{R}_{\geq 0} \to \mathbb{R}$ with $\Phi(1) = 0$, the $\Phi$-divergence between probability distributions $\mu$ and $\nu$ such that $\mu \ll \nu$ is defined by

$$D_\Phi(\mu \parallel \nu) := \mathbb{E}_\nu\left[\Phi\left(\frac{\mu}{\nu}\right)\right], \tag{1}$$

---

1. We use "distance" loosely here to refer to metrics such as Wasserstein metric, or divergences such as KL divergence.
2. Also commonly known as $f$-divergences in the literature.

and it is $+\infty$ if $\mu \not\ll \nu$. For example, the KL divergence or relative entropy corresponds to $\Phi(x) = x \log x$, chi-squared divergence corresponds to $\Phi(x) = (x-1)^2$, TV distance corresponds to $\Phi(x) = \frac{1}{2}|x-1|$, and squared-Hellinger distance corresponds to $\Phi(x) = \frac{1}{2}(\sqrt{x}-1)^2$. In general, $\Phi$-divergences do not satisfy a triangle inequality, but some examples which do are the TV distance and Marton's divergence (Sason and Verdú, 2016, (72)). Further examples of $\Phi$-divergences can be found in Table 1 in Appendix A. The family of $\Phi$-divergences have found profound applications from hypothesis and distribution testing (Pensia et al., 2024; Györfi and Vajda, 2002; Gretton and Györfi, 2008), and neuroscience (Nemenman et al., 2004; Belitski et al., 2008), to reinforcement learning (Ho et al., 2022; Panaganti et al., 2024). We study $\Phi$-divergences in the context of mixing time for Markov chains, and extend the analysis from KL divergence[3] to $\Phi$-divergence.

We study the mixing time of Markov chains in $\Phi$-divergence under the assumption that their stationary distribution satisfy a $\Phi$-Sobolev inequality. These inequalities include as special cases popular isoperimetric inequalities such as the log-Sobolev inequality (LSI) and Poincaré inequality, and are defined as follows.

**Definition 1** *A probability distribution $\nu$ satisfies a $\Phi$-Sobolev inequality ($\Phi$SI) with constant $\alpha > 0$ if for all probability distributions $\mu \ll \nu$, we have*

$$2\alpha\, \mathsf{D}_\Phi(\mu \,\|\, \nu) \leq \mathsf{FI}_\Phi(\mu \,\|\, \nu), \tag{2}$$

*where $\mathsf{D}_\Phi(\mu \,\|\, \nu)$ is the $\Phi$-divergence defined in (1), and $\mathsf{FI}_\Phi(\mu \,\|\, \nu)$ is the $\Phi$-Fisher information defined by*

$$\mathsf{FI}_\Phi(\mu \,\|\, \nu) := \underset{\nu}{\mathbb{E}}\left[\left\|\nabla\frac{\mu}{\nu}\right\|^2 \Phi''\left(\frac{\mu}{\nu}\right)\right].$$

*We define the $\Phi$-Sobolev constant of $\nu$ to be the optimal (largest) constant $\alpha$ such that the above inequality holds:*

$$\alpha_{\Phi\mathsf{SI}}(\nu) := \inf_\mu \frac{\mathsf{FI}_\Phi(\mu \,\|\, \nu)}{2\mathsf{D}_\Phi(\mu \,\|\, \nu)} \tag{3}$$

*where the infimum is taken over all probability distributions $\mu$ with $0 < \mathsf{D}_\Phi(\mu \,\|\, \nu) < \infty$.*

The inequality (2) is equivalent to saying that for all smooth functions $g : \mathbb{R}^d \to \mathbb{R}_{\geq 0}$ with $\mathbb{E}_\nu[g] = 1$,

$$2\alpha\, \mathsf{Ent}_\Phi^\nu(g) \leq \mathcal{E}_\Phi^\nu(g), \tag{4}$$

where

$$\mathsf{Ent}_\Phi^\nu(g) := \underset{\nu}{\mathbb{E}}[\Phi(g)] - \Phi(\underset{\nu}{\mathbb{E}}[g]) \quad \text{and} \quad \mathcal{E}_\Phi^\nu(g) := \underset{\nu}{\mathbb{E}}\left[\|\nabla g\|^2 \Phi''(g)\right].$$

To see this, take $g$ to be the density function of $\mu$ with respect to $\nu$.

For $\Phi(x) = x \log x$, the $\Phi$-divergence is the KL divergence and the $\Phi$-Sobolev inequality is the log-Sobolev inequality. For $\Phi(x) = (x-1)^2$, the $\Phi$-divergence is the chi-squared divergence and the $\Phi$-Sobolev inequality is the Poincaré inequality. Further examples are mentioned in Table 1 in Appendix A. The Poincaré inequality is the weakest $\Phi$-Sobolev inequality in that it is implied by any other $\Phi$-Sobolev inequality (Chafaï, 2004, Section 2.2). This extends the more well-known

---

3. And other specific analyses such as in chi-squared divergence.

statement that the log-Sobolev inequality implies the Poincaré inequality. Additionally, one can start from a Φ-Sobolev inequality and show that it satisfies a Φ-Sobolev inequality for $\Phi(x) = x^p - 1$ for $p \in (1, 2]$ but without a tight constant (Chafaï, 2004, Section 2.2). This class of Φ-Sobolev inequalities are related to Beckner inequalities (Bakry et al., 2014, Section 7.6.2).

We discuss how the Φ-Sobolev constant of a distribution evolves along various operations such as convolutions and pushforwards in Section 2.1. These properties will be crucial when analyzing mixing time. We now introduce the Markov chains we will study.

### 1.1. Langevin dynamics

The Langevin dynamics to sample from $\nu \propto \exp(-f)$ on $\mathbb{R}^d$ is the following stochastic differential equation (SDE):

$$\mathrm{d}X_t = -\nabla f(X_t)\,\mathrm{d}t + \sqrt{2}\,\mathrm{d}W_t\,, \tag{5}$$

where $W_t$ is the standard Brownian motion on $\mathbb{R}^d$. The Langevin dynamics admits $\nu$ as the stationary or invariant distribution, and hence is a natural process to study for sampling (Bakry et al., 2014). It also has the natural optimization interpretation as the gradient flow to minimize KL divergence in the space of probability distributions over $\mathbb{R}^d$ with the Wasserstein $W_2$ metric, see (Jordan et al., 1998; Wibisono, 2018).

The Langevin dynamics is a continuous-time Markov process, and it needs to be discretized in time in order to implement in practice. We will focus on two discretizations of the Langevin dynamics – the Unadjusted Langevin Algorithm (ULA), which has been well-studied in (Roberts and Tweedie, 1996; Roberts and Rosenthal, 1998; Dalalyan, 2017; Cheng and Bartlett, 2018; Vempala and Wibisono, 2019; Chewi et al., 2022a), and the Proximal Sampler, which has been studied in (Lee et al., 2021; Chen et al., 2022; Yuan et al., 2023; Kook et al., 2024; Kook and Zhang, 2025).

The convergence of Φ-divergence along Langevin dynamics is easy to establish (Corollary 9) and the properties of Φ-divergence along continuous-time dynamics are well-studied (Chafaï, 2004; Dolbeault and Li, 2018; Achleitner et al., 2015). We study the convergence of Φ-divergence for discrete-time samplers in Theorems 2 and 3.

### 1.2. Unadjusted Langevin algorithm

The Unadjusted Langevin Algorithm (ULA) is a simple discretization of the Langevin dynamics (5) and is given by the following update

$$X_{k+1} = X_k - \eta\nabla f(X_k) + \sqrt{2\eta}Z_k\,, \tag{6}$$

where $Z_k \sim \mathcal{N}(0, I)$ and $\eta > 0$ is the stepsize. It is well-known (Roberts and Tweedie, 1996) that the ULA is a biased discretization, which means that it admits as its stationary distribution $\nu^\eta \neq \nu$ for all $\eta > 0$; furthermore, as $\eta k \to t$ and $\eta \to 0$, $\nu^\eta \to \nu$ and ULA (6) recovers the Langevin dynamics (5). The ULA has been widely studied, and we now have guarantees on its mixing time in numerous settings: Dalalyan (2017) studies the mixing time to $\nu$ in Wasserstein $W_2$ distance under $\nu$ being strongly log-concave; Cheng and Bartlett (2018) study the mixing time to $\nu$ in KL divergence under $\nu$ being strongly log-concave; Vempala and Wibisono (2019) study mixing to $\nu$ in KL divergence under an LSI assumption on $\nu$, and also study mixing to the biased limit $\nu^\eta$ in Rényi divergence under an LSI assumption on $\nu^\eta$; Altschuler and Talwar (2023) study mixing to $\nu^\eta$ under a strong log-concavity assumption on $\nu$; and Chewi et al. (2022a) study mixing to $\nu$ under modified

LSI assumptions on $\nu$. We refer the reader to Chewi (2024) for a comprehensive overview of recent mixing time results.

A probability distribution $\nu \propto \exp(-f)$ is said to be $L > 0$ smooth if $-LI \preceq \nabla^2 f \preceq LI$, and all of the aforementioned works make smoothness assumptions on the target distribution $\nu$. This is common for discrete-time analysis. In Theorem 2, we show mixing guarantees for ULA in $\Phi$-divergence for smooth target distributions $\nu$ and under the assumption that the stationary distribution $\nu^\eta$ of ULA satisfies the corresponding $\Phi$-Sobolev inequality (Definition 1). To the best of our knowledge, we are the first to study mixing time guarantees in any $\Phi$-divergence for ULA. We prove Theorem 2 in Section 3.2.

**Theorem 2** *Suppose the stationary distribution $\nu^\eta$ of ULA satisfies a $\Phi$-Sobolev inequality with optimal constant $\alpha > 0$, and $\nu$ is $L$-smooth for some $0 < \alpha \leq L < \infty$. Let $X_k \sim \rho_k$ evolve following ULA (6) with step size $0 < \eta \leq 1/L$ from $X_0 \sim \rho_0$. Then for all $k \geq 1$,*

$$\mathsf{D}_\Phi(\rho_k \parallel \nu^\eta) \leq \left(1 + \frac{2\alpha\eta}{(1+\eta L)^2}\right)^{-k} \mathsf{D}_\Phi(\rho_0 \parallel \nu^\eta). \tag{7}$$

As $\eta k \to t$ and $\eta \to 0$, Theorem 2 recovers the correct $\exp(-2\alpha t)$ convergence rate for Langevin dynamics under a $\Phi$-Sobolev inequality (see Corollary 9 in Section 2.1). We show that the rate of Theorem 2 is tight for KL divergence via an explicit calculation for the case when $\nu$ is a Gaussian, see Proposition 17 in Appendix G.1.

A common setting in which $\nu^\eta$ satisfies a $\Phi$-Sobolev inequality is when $\nu$ is strongly log-concave (see Lemma 13 in Section 3.3). To the best of our knowledge, properties of the limiting distribution $\nu^\eta$ under assumptions on $\nu$ which are weaker than strong log-concavity remain unknown. Similar to how log-Sobolev and Poincaré inequalities are stable to bounded perturbations (Holley and Stroock, 1987), $\Phi$-Sobolev inequalities also enjoy similar properties (Chafaï, 2004, Section 3.3). Therefore, an interesting question one can ask is if the stationary distribution $\nu^\eta$ continues to satisfy a $\Phi$-Sobolev inequality when $\nu$ undergoes a suitable perturbation; we leave this for future work.

Theorem 2 shows convergence to the stationary distribution $\nu^\eta$. The bias $\mathsf{D}_\Phi(\nu^\eta \parallel \nu)$ between the stationary distribution and the target distribution depends on the choice of $\Phi$. For example, for the Ornstein-Uhlenbeck process where $\nu = \mathcal{N}(0, \frac{1}{\alpha}I)$, we have $\nu^\eta = \mathcal{N}(0, \frac{2}{\alpha(2-\eta\alpha)}I)$ (see Appendix G.1). In this case, the bias has linear dependence in the dimension $d$ for KL divergence ($\Phi(x) = x \log x$) and exponential dependence in $d$ for chi-squared divergence ($\Phi(x) = (x-1)^2$).

### 1.3. Proximal Sampler

While the simplicity of ULA is appealing, its biased limiting distribution makes it less ideal for applications requiring sampling with high accuracy. The Proximal Sampler (Lee et al., 2021; Chen et al., 2022) is an alternative discretization that addresses this shortcoming. Given the close connections between optimization and sampling, along with the modern interpretation of sampling as optimization in the space of distributions (Jordan et al., 1998; Wibisono, 2018), the Proximal Sampler as introduced in Lee et al. (2021) can be seen to be a sampling analogue of the proximal point algorithm in optimization (Martinet, 1970; Rockafellar, 1976). The Proximal Sampler was analyzed in great generality in Chen et al. (2022), who show mixing guarantees in KL divergence, Rényi divergence, chi-square divergence, and $W_2$ distance, under isoperimetry or log-concavity assumptions on $\nu$. In Theorem 3, we show mixing guarantees for the Proximal Sampler in $\Phi$-divergence under

the assumption that $\nu$ satisfies the corresponding $\Phi$-Sobolev inequality (Definition 1). Our results extend the mixing time of the Proximal Sampler to $\Phi$-divergence.

The Proximal Sampler considers an augmented $(X, Y)$ space $\mathbb{R}^d \times \mathbb{R}^d$ and performs Gibbs sampling on the joint space. We will use the appropriate superscripts to denote probability distributions on their respective spaces.[4] Let the target distribution we wish to sample from be $\nu^X \propto \exp(-f)$ on $\mathbb{R}^d$. The joint target distribution on $\mathbb{R}^d \times \mathbb{R}^d$ is defined as:

$$\nu^{XY}(x,y) \propto \exp\left(-f(x) - \frac{\|x-y\|^2}{2\eta}\right), \tag{8}$$

for step size $\eta > 0$. The Proximal Sampler, initialized from $X_0 \sim \rho_0^X$, is the following two-step algorithm:

**Step 1 (forward step):** Sample $Y_k \mid X_k \sim \nu^{Y|X=X_k} = \mathcal{N}(X_k, \eta I)$

**Step 2 (backward step):** Sample $X_{k+1} \mid Y_k \sim \nu^{X|Y=Y_k}$

$$\tag{9}$$

Note that $\nu^{XY}$ has the desired target distribution $\nu^X$ as the $X$ marginal, and that the $Y$ marginal $\nu^Y$ is a smoothed version of $\nu^X$, i.e., $\nu^Y = \nu^X * \mathcal{N}(0, \eta I)$. The forward step of the algorithm is easy to implement as it corresponds to drawing a Gaussian random variable. For the backward step, implementation is possible given access to a *Restricted Gaussian Oracle* (RGO). A RGO is an oracle that, given any $y \in \mathbb{R}^d$, outputs a sample from $\nu^{X|Y=y}$, i.e. from

$$\nu^{X|Y}(x \mid y) \propto_x \exp\left(-f(x) - \frac{\|x-y\|^2}{2\eta}\right).$$

Similar to Lee et al. (2021); Chen et al. (2022), in our main result for the Proximal Sampler (Theorem 3), we consider an ideal implementation of the sampler where we have exact access to a RGO. Specific cases where the RGO can be implemented efficiently are discussed in Lee et al. (2021), and improved implementations of the RGO along with analysis of the Proximal Sampler with inexact RGO implementations have been an active area of research (Fan et al., 2023; Liang and Chen, 2022; Altschuler and Chewi, 2024). Fan et al. (2023) show an implementation via approximate rejection sampling and Altschuler and Chewi (2024) show an approach based on the Metropolis-adjusted Langevin algorithm.

We mention a basic approach to implement the RGO via rejection sampling in Appendix B. We also mention the oracle complexity (in this case, the expected number of calls to the first order oracle of $f$) when sampling using the Proximal Sampler with the rejection sampling based RGO implementation in Corollary 4.

When both steps are implemented exactly (i.e. given access to an exact RGO), the Proximal Sampler corresponds to Gibbs sampling from the stationary distribution $\nu^{XY}$, and the algorithm is therefore unbiased. Step 1 is referred to as the forward step as it corresponds to evolving along the (forward) heat flow, and step 2 is referred to as the backward step is it corresponds to backward heat flow; this perspective is further made clear in Sections 4.1 and 4.2. We now state the main theorem describing the mixing time of the Proximal Sampler in $\Phi$-divergence. We prove Theorem 3 in Section 4.3.

---

4. For example, $\rho^{XY}$ denotes a distribution on the joint space, $\rho^X$ refers to a distribution on the $X$ space, and $\rho^{Y|X=x}$ denotes a conditional distribution supported on the $Y$ space.

**Theorem 3** *Suppose $\nu^X$ satisfies a $\Phi$-Sobolev inequality with optimal constant $\alpha$. Let $X_k \sim \rho_k^X$ evolve along the Proximal Sampler* (9) *with step size $\eta > 0$ from $X_0 \sim \rho_0^X$. Then for all $k \geq 1$,*

$$\mathsf{D}_\Phi(\rho_k^X \parallel \nu^X) \leq \frac{\mathsf{D}_\Phi(\rho_0^X \parallel \nu^X)}{(1 + \alpha\eta)^{2k}} .$$

We show the tightness of Theorem 3 in KL divergence by doing an explicit calculation for the case when $\nu$ is Gaussian, see Proposition 18 in Appendix G.2. The setting of Theorem 3 includes popular mixing time results as special cases. For example, it directly implies the mixing in KL divergence under a log-Sobolev inequality, and mixing in chi-squared divergence under a Poincaré inequality. Under a strong log-concavity assumption on $\nu$, it implies mixing in all $\Phi$-divergences. $\Phi$-Sobolev inequalities and $\Phi$-divergences for different choices of $\Phi$ are mentioned in Table 1 in Appendix A.

Theorem 3 considers the ideal implementation of the Proximal Sampler where we assume exact access to the RGO. Combined with an RGO implementation via rejection sampling, which requires an additional smoothness assumption on $f$, Theorem 3 yields the following corollary.

**Corollary 4** *Suppose $\nu^X$ satisfies a $\Phi$-Sobolev inequality with optimal constant $\alpha$ and is $L$-smooth. Then for any $\epsilon > 0$, the Proximal Sampler with $\eta \asymp \frac{1}{Ld}$ and with rejection sampling based RGO implementation (as described in Appendix B) outputs $X_k \sim \rho_k^X$ with $\mathsf{D}_\Phi(\rho_k^X \parallel \nu^X) \leq \epsilon$ whenever $k \geq \frac{Ld}{2\alpha} \log \frac{\mathsf{D}_\Phi(\rho_0^X \parallel \nu^X)}{\epsilon}$. The expected number of oracle calls to $f$ is $\mathcal{O}\big(\frac{Ld}{\alpha} \log \frac{\mathsf{D}_\Phi(\rho_0^X \parallel \nu^X)}{\epsilon}\big)$.*

## 1.4. Related work

Our main results, Theorems 2 and 3, study the mixing time in $\Phi$-divergence of the ULA (6) and Proximal Sampler (9) respectively. We give an overview of the algorithms along with prior mixing time results in Sections 1.2 and 1.3, and refer the reader to those sections for the corresponding references. We discuss other related works here; see also Chewi (2024) for a comprehensive overview of Langevin-based samplers.

Both the ULA (6) and Proximal Sampler (9) have the Langevin dynamics (5) as the limiting dynamics as $\eta \to 0$, and the convergence of $\Phi$-divergence along the Langevin dynamics is well-known (see Corollary 9, and see also Chafaï (2004); Dolbeault and Li (2018); Achleitner et al. (2015) for a general discussion for diffusions). To study the convergence for discrete-time algorithms, our main tool is that of Strong Data Processing Inequalities (SDPIs) (see Section 2.2). SDPIs are Markov chain-dependent strengthenings of the data processing inequality, and are a fundamental concept in information theory. For a comprehensive overview of SDPIs, we refer the reader to Polyanskiy and Wu (2024, Chapter 33). Just as data processing inequalities hold in many different metrics, so do SDPIs. Raginsky (2016) study SDPIs in $\Phi$-divergence for discrete-space Markov chains, and du Pin Calmon et al. (2017); Polyanskiy and Wu (2017) study SDPIs in lesser generality than $\Phi$-divergences, but for continuous-space chains and networks.

SDPI-inspired techniques have been used in prior works studying the mixing time of Langevin based algorithms, but most works use them implicitly, and to the best of our knowledge, none study $\Phi$-divergences in general. Vempala and Wibisono (2019) use them to study the convergence of ULA to $\nu^\eta$ in Rényi divergence, Chen et al. (2022) use them for the Proximal Sampler, Yuan et al. (2023) mention SDPIs and use them for the Proximal Sampler on graphs, Kook et al. (2024); Kook and Zhang (2025) use them for the constrained Proximal Sampler. We make the SDPI-based approach

explicit (Section 2.2) and use it for Φ-divergence. It should also be noted that Rényi divergence is not a Φ-divergence, but it is a simple transformation of a Φ-divergence, so SDPI-based analyses have also been used to show mixing time in Rényi divergence: Vempala and Wibisono (2019) present mixing guarantees in Rényi divergence via SDPI for the ULA, and Chen et al. (2022) for the Proximal Sampler.

Bounding the contraction coefficient (Definition 10) is the key step in showing a Markov chain satisfies a SDPI, and the main idea used to bound the contraction coefficient for Langevin-based Markov chains is by taking the time derivative of the divergence along two simultaneous stochastic processes (Lemma 8). This idea traces back to de Bruijn's identity (Stam, 1959), and similar methods have been used well beyond information theory, for example to study diffusions and diffusion models (Chafaï, 2004; Albergo et al., 2023; Vempala and Wibisono, 2019; Kook et al., 2024).

**Organization** We go over the necessary background material in Section 2, and then discuss the convergence of Φ-divergence along ULA in Section 3, and along the Proximal Sampler in Section 4. We summarize and discuss open questions in Section 5 to conclude.

## 2. Preliminaries

A distribution $\nu \propto \exp(-f)$ on $\mathbb{R}^d$ with a twice-differentiable potential function $f$ is $\alpha$-strongly log-concave for some $\alpha > 0$ if $\alpha I \preceq \nabla^2 f$, and is $L$-smooth for some $L > 0$ if $-LI \preceq \nabla^2 f \preceq LI$. When $\alpha = 0$, we call $\nu$ (weakly) log-concave. Throughout, we take $\Phi : \mathbb{R}_{\geq 0} \to \mathbb{R}$ to be a twice-differentiable strictly convex function with $\Phi(1) = 0$. Whenever we refer to any probability distribution, we always take it to be a member of $\mathcal{P}_{2,ac}(\mathbb{R}^d)$, i.e. the set of probability distributions on $\mathbb{R}^d$ which are absolutely continuous with respect to Lebesgue measure and have finite second moment. We also refer to distributions via their densities with respect to Lebesgue measure. We denote $\mathcal{N}_t$ as shorthand for $\mathcal{N}(0, tI)$ where $t > 0$, and use $\mathcal{N}(\mu, \Sigma)$ to refer to a Gaussian distribution with mean $\mu \in \mathbb{R}^d$ and positive-definite covariance matrix $\Sigma \in \mathbb{R}^{d \times d}$.

### 2.1. Φ-divergences, Φ-Sobolev inequalities, and their properties

Recall the Φ-divergence between probability distributions is defined in (1). To study the mixing time of the ULA (Theorem 2) and Proximal Sampler (Theorem 3) in Φ-divergence, we assume a Φ-Sobolev inequality assumption on the stationary distribution (Definition 1).

Our proofs of Theorems 2 and 3 are based on SDPIs (Section 2.2), and core to this SDPI-based proof strategy will be analyzing the change of the Φ-Sobolev constant along various operations such as convolution and pushforward. These operations arise by interpreting the Markov chains (6) and (9) as updates in the space of distributions, and will be described in Sections 3 and 4 respectively. The evolution of the Φ-Sobolev constant along these operations is classical (Chafaï, 2004) and we discuss them now.

The following lemma tells us how the ΦSI constant evolves along a Lipschitz pushforward map.

**Lemma 5** *(Chafaï, 2004, Remark 7) Assume $\nu$ satisfy Φ-Sobolev inequality with optimal constant $\alpha_{\Phi \mathsf{SI}}(\nu)$. Let $T : \mathbb{R}^d \to \mathbb{R}^d$ be a $\gamma$-Lipschitz map. Then, the pushforward $\tilde{\nu} = T_{\#}\nu$ satisfies Φ-Sobolev inequality with optimal constant*

$$\alpha_{\Phi \mathsf{SI}}(\tilde{\nu}) \geq \frac{\alpha_{\Phi \mathsf{SI}}(\nu)}{\gamma^2}.$$

The next lemma describes the change of the $\Phi\mathsf{SI}$ constant after convolution.

**Lemma 6** *([Chafaï](), [2004](), Corollary 3.1) Assume $\mu$ and $\nu$ satisfy $\Phi\mathsf{SI}$ with optimal constants $\alpha_{\Phi\mathsf{SI}}(\mu)$ and $\alpha_{\Phi\mathsf{SI}}(\nu)$, respectively. Then the convolution $\mu * \nu$ satisfies $\Phi\mathsf{SI}$ with constant*

$$\frac{1}{\alpha_{\Phi\mathsf{SI}}(\mu * \nu)} \leq \frac{1}{\alpha_{\Phi\mathsf{SI}}(\mu)} + \frac{1}{\alpha_{\Phi\mathsf{SI}}(\nu)}.$$

The following lemma tells us that when $\nu$ is $\alpha$-strongly log-concave, it also satisfies $\Phi$-Sobolev inequality with the same constant.

**Lemma 7** *([Chafaï](), [2004](), Corollary 2.1) If $\nu$ is $\alpha$-strongly log-concave for some $\alpha > 0$, then $\nu$ satisfies $\Phi\mathsf{SI}$ with constant*

$$\alpha_{\Phi\mathsf{SI}}(\nu) \geq \alpha.$$

We conclude this subsection by describing the rate of change of $\Phi$-divergence along simultaneous evolutions of the same SDE. The following lemma will be crucial in the SDPI-based approach. The same lemma described in terms of Markov semigroup theory can be found in [Chewi]() ([2024](), Theorem 8.3.1). We prove Lemma 8 in Appendix C.

**Lemma 8** *Suppose $X_t \sim \mu_t$ and $X_t \sim \nu_t$ with initial conditions $\mu_0$ and $\nu_0$ are two solutions of the following SDE:*

$$\mathrm{d}X_t = b_t(X_t)\,\mathrm{d}t + \sqrt{2c}\,\mathrm{d}W_t, \tag{10}$$

*where $b_t : \mathbb{R}^d \to \mathbb{R}^d$ is a time-varying drift function, $c$ is a positive constant, and $W_t$ is the standard Brownian motion on $\mathbb{R}^d$. Then for all $t \geq 0$,*

$$\frac{\mathrm{d}}{\mathrm{d}t}\mathsf{D}_\Phi(\mu_t \,\|\, \nu_t) = -c\,\mathsf{FI}_\Phi(\mu_t \,\|\, \nu_t).$$

As mentioned in Section 1.4, when (10) is taken to be Brownian motion (i.e. the drift $b_t = 0$), $\nu_t$ is fixed to be the Lebesgue measure, and $\Phi(x) = x \log x$, Lemma 8 corresponds to de Bruijn's identity ([Stam](), [1959]()). The SDE (10) is more general than the Langevin dynamics (5) as it includes a time-varying drift function; this is required to study the Proximal Sampler, since as we will see in Section 4, the backward step of the Proximal Sampler corresponds to an SDE with time-varying drift.

As an easy consequence of Lemma 8, we have the following exponential convergence of $\Phi$-divergence for the Langevin dynamics when $\nu$ satisfies a $\Phi$-Sobolev inequality.

**Corollary 9** *Suppose $X_t \sim \rho_t$ evolves along the Langevin dynamics (5) to sample from $\nu \propto \exp(-f)$, and let $\nu$ satisfy a $\Phi$-Sobolev inequality with optimal constant $\alpha > 0$. Then,*

$$\mathsf{D}_\Phi(\rho_t \,\|\, \nu) \leq e^{-2\alpha t}\mathsf{D}_\Phi(\rho_0 \,\|\, \nu).$$

**Proof** Applying Lemma 8 (with $\mu_t = \rho_t$ and $\nu_t = \nu$) along with the $\Phi$-Sobolev inequality of $\nu$ yields,

$$\frac{\mathrm{d}}{\mathrm{d}t}\mathsf{D}_\Phi(\mu_t \,\|\, \nu) = -\mathsf{FI}_\Phi(\mu_t \,\|\, \nu) \leq -2\alpha\mathsf{D}_\Phi(\mu_t \,\|\, \nu).$$

Integrating the differential inequality from $0$ to $t$ using Grönwall's lemma completes the proof. ∎

## 2.2. Strong data processing inequalities

We now discuss Strong Data Processing Inequalities (SDPIs), which will be our primary proof strategy. For any Φ-divergence, the data processing inequality states that for any two distributions $\mu$ and $\nu$, and for any Markov kernel $\mathbf{Q}$, $\mathsf{D}_\Phi(\mu\mathbf{Q} \,\|\, \nu\mathbf{Q}) \leq \mathsf{D}_\Phi(\mu \,\|\, \nu)$, provided all quantities are well-defined.[5] *Strong* data processing inequalities check if the inequality is strict, and quantify the decrease (Polyanskiy and Wu, 2024, 2017; Raginsky, 2016). They do so by fixing the second input distribution (i.e. $\nu$) and then varying the first distribution (i.e. $\mu$), to see the worst case ratio $\frac{\mathsf{D}_\Phi(\mu\mathbf{Q}\|\nu\mathbf{Q})}{\mathsf{D}_\Phi(\mu\|\nu)}$. The ratio is called the *contraction coefficient*, and when it is strictly less than 1 for all valid distributions $\mu$, we say that $(\mathbf{Q}, \nu)$ satisfies a strong data processing inequality.

Next, we define the contraction coefficient in Definition 10 and define SDPIs in Φ-divergence in Definition 11.

**Definition 10** *Let $\nu$ be a probability distribution, $\mathbf{Q}$ be a Markov kernel, and $\mathsf{D}_\Phi$ be a Φ-divergence. Then the contraction coefficient $\varepsilon_{\mathsf{D}_\Phi}$ is defined as follows*

$$\varepsilon_{\mathsf{D}_\Phi}(\mathbf{Q}, \nu) := \sup_{\rho \,:\, 0 < \mathsf{D}_\Phi(\rho\|\nu) < \infty} \frac{\mathsf{D}_\Phi(\rho\mathbf{Q} \,\|\, \nu\mathbf{Q})}{\mathsf{D}_\Phi(\rho \,\|\, \nu)}. \tag{11}$$

**Definition 11** *Let $\nu$ be a probability distribution, $\mathbf{Q}$ be a Markov kernel, and $\mathsf{D}_\Phi$ be a Φ-divergence. Further define $\varepsilon_{\mathsf{D}_\Phi}(\mathbf{Q}, \nu)$ as in* (11). *Then we say that $(\mathbf{Q}, \nu)$ satisfies a strong data processing inequality in Φ-divergence when $\varepsilon_{\mathsf{D}_\Phi}(\mathbf{Q}, \nu) < 1$. In particular, we have,*

$$\mathsf{D}_\Phi(\mu\mathbf{Q} \,\|\, \nu\mathbf{Q}) \leq \varepsilon_{\mathsf{D}_\Phi}(\mathbf{Q}, \nu)\mathsf{D}_\Phi(\mu \,\|\, \nu), \tag{12}$$

*where $\mu$ is any distribution such that $\mathsf{D}_\Phi(\mu \,\|\, \nu) < \infty$.*

As a direct consequence of Definition 11, we can see that if $\nu$ is invariant for $\mathbf{Q}$, then for any $\mu$,

$$\mathsf{D}_\Phi(\mu\mathbf{Q}^k \,\|\, \nu) \leq \varepsilon_{\mathsf{D}_\Phi}(\mathbf{Q}, \nu)^k \,\mathsf{D}_\Phi(\mu \,\|\, \nu). \tag{13}$$

Hence, if we desire to show a mixing time result for Markov chain $\mathbf{Q}$ with invariant distribution $\nu$, then we require a bound on $\varepsilon_{\mathsf{D}_\Phi}(\mathbf{Q}, \nu)$ which is strictly less than 1.

## 3. Convergence Along ULA

Recall the ULA update (6) for sampling from $\nu \propto \exp(-f)$. Defining $\rho_k := \mathsf{law}(X_k)$, the update (6) can be seen as the following update in distribution:

$$\rho_{k+1} = (\mathsf{id} - \eta\nabla f)_{\#}\rho_k * \mathcal{N}(0, 2\eta I). \tag{14}$$

Denote the Markov kernel of ULA as $\mathbf{P}$ (i.e. $\rho_{k+1} = \rho_k\mathbf{P} = \rho_0\mathbf{P}^{k+1}$), and define $F \colon \mathbb{R}^d \to \mathbb{R}^d$ by $F(x) = x - \eta\nabla f(x)$. Recall from Section 1.2 that ULA has a biased stationary distribution $\nu^\eta$. Then as per (13), in order to obtain a mixing time result, our goal is to bound $\varepsilon_{\mathsf{D}_\Phi}(\mathbf{P}, \nu^\eta)$. We now provide a proof outline for this.

---

5. For an example of the Markov kernel notation, and its behaviour as on operator on distributions, see (14) and the subsequent text.

### 3.1. Proof outline

We wish to bound $\varepsilon_{\mathsf{D}_\Phi}(\mathbf{P}, \nu^\eta)$. To that end, let $\mu$ be an arbitrary probability distribution and recall by Definition 10 that we want to control

$$\frac{\mathsf{D}_\Phi(\mu\mathbf{P} \parallel \nu^\eta\mathbf{P})}{\mathsf{D}_\Phi(\mu \parallel \nu^\eta)} = \frac{\mathsf{D}_\Phi(F_\#\mu * \mathcal{N}(0, 2\eta I) \parallel F_\#\nu^\eta * \mathcal{N}(0, 2\eta I))}{\mathsf{D}_\Phi(\mu \parallel \nu^\eta)}.$$

Suppose that $F$ is a bijective map. Then, as $\Phi$-divergence is invariant to simultaneous bijective deterministic maps, we have $\mathsf{D}_\Phi(\mu \parallel \nu^\eta) = \mathsf{D}_\Phi(F_\#\mu \parallel F_\#\nu^\eta)$.[6] Therefore, under the assumption that $F$ is bijective, the quantity we wish to bound is

$$\frac{\mathsf{D}_\Phi(F_\#\mu * \mathcal{N}(0, 2\eta I) \parallel F_\#\nu^\eta * \mathcal{N}(0, 2\eta I))}{\mathsf{D}_\Phi(F_\#\mu \parallel F_\#\nu^\eta)}.$$

Denoting $F_\#\mu * \mathcal{N}(0, tI) = \mu_t$ and $F_\#\nu^\eta * \mathcal{N}(0, tI) = \nu_t$, we want to bound

$$\frac{\mathsf{D}_\Phi(\mu_{2\eta} \parallel \nu_{2\eta})}{\mathsf{D}_\Phi(\mu_0 \parallel \nu_0)}.$$

As convolving with a Gaussian can be viewed as the solution to the heat equation, this quantity can be bounded as a consequence of Lemma 8 (for the Brownian motion SDE, i.e., for (10) with $b_t = 0$ and $c = \frac{1}{2}$) under the assumption that $\nu_t$ satisfies a $\Phi$-Sobolev inequality for $t \in [0, 2\eta]$. The final expression obtained is independent of $\mu$, and therefore this provides a bound on $\varepsilon_{\mathsf{D}_\Phi}(\mathbf{P}, \nu^\eta)$.

This is the outline we follow and the assumptions stated in Theorem 2 are to ensure that the assumptions mentioned in the proof sketch go through.

### 3.2. Proof of Theorem 2

In this section we prove Theorem 2. The key lemma for doing so is the following. The proof of Lemma 12 follows the outline mentioned in Section 3.1 and is in Appendix D.

**Lemma 12**  *Let $\mathbf{P}$ denote the ULA Markov kernel with update (14). Suppose $\nu^\eta$ satisfies a $\Phi$-Sobolev inequality with optimal constant $\alpha$, $\nu$ is $L$-smooth, and $\eta \leq \frac{1}{L}$. Then,*

$$\varepsilon_{\mathsf{D}_\Phi}(\mathbf{P}, \nu^\eta) \leq \frac{(1 + \eta L)^2}{(1 + \eta L)^2 + 2\alpha\eta}.$$

Lemma 12 gives a bound on the contraction coefficient for ULA, with which we can prove Theorem 2.

**Proof of Theorem 2** By an application of Definition 11 and by noting that $\nu^\eta$ is stationary for $\mathbf{P}$, we have that,

$$\mathsf{D}_\Phi(\rho_k \parallel \nu^\eta) \leq \varepsilon_{\mathsf{D}_\Phi}(\mathbf{P}, \nu^\eta)^k \, \mathsf{D}_\Phi(\rho_0 \parallel \nu^\eta).$$

Using Lemma 12, we get that,

$$\mathsf{D}_\Phi(\rho_k \parallel \nu^\eta) \leq \left(\frac{(1 + \eta L)^2}{(1 + \eta L)^2 + 2\alpha\eta}\right)^k \mathsf{D}_\Phi(\rho_0 \parallel \nu^\eta) = \left(1 + \frac{2\alpha\eta}{(1 + \eta L)^2}\right)^{-k} \mathsf{D}_\Phi(\rho_0 \parallel \nu^\eta).$$

---

6. This can be seen as a consequence of data processing inequality, i.e. for any $\mu$ and $\rho$ and bijective map $F$, $\mathsf{D}_\Phi(\mu\|\rho) \geq \mathsf{D}_\Phi(F_\#\mu \parallel F_\#\rho) \geq \mathsf{D}_\Phi(F_\#^{-1}F_\#\mu \parallel F_\#^{-1}F_\#\rho) = \mathsf{D}_\Phi(\mu \parallel \rho)$, hence all the inequalities must be equalities.

An explicit calculation for the Ornstein-Uhlenbeck process implying the tightness of Theorem 2 for KL divergence can be found in Proposition 17 in Appendix G.1.

### 3.3. Property of the biased limit

We conclude this section by showing that the biased limit satisfies a Φ-Sobolev inequality under strong log-concavity of $\nu$. We prove Lemma 13 in Appendix E.

**Lemma 13** *Suppose $\nu \propto \exp(-f)$ is $\alpha$-strongly log-concave and $L$-smooth. Consider the ULA (6) to sample from $\nu$ with step size $\eta \leq \frac{1}{L}$. Then the biased limit $\nu^\eta$ satisfies:*

$$\alpha_{\Phi\mathsf{SI}}(\nu^\eta) \geq \frac{\alpha}{2}\,.$$

## 4. Convergence Along Proximal Sampler

Recall the Proximal Sampler from Section 1.3 with update given by (9), with the forward and backward steps. We will denote the Proximal Sampler as $\mathbf{P}_{\mathrm{prox}} = \mathbf{P}^+_{\mathrm{prox}}\mathbf{P}^-_{\mathrm{prox}}$ where $\mathbf{P}^+_{\mathrm{prox}}$ corresponds to the forward step and $\mathbf{P}^-_{\mathrm{prox}}$ to the backward step. Each step of the Proximal Sampler is an update on $\mathbb{R}^d$, and so this perspective via composition of Markov kernels is valid. In terms of notation, we have $\rho^X_k := \mathsf{law}(X_k)$, $\rho^Y_k := \mathsf{law}(Y_k)$, and therefore $\rho^X_k \mathbf{P}_{\mathrm{prox}} = \rho^X_{k+1}$, $\rho^X_k \mathbf{P}^+_{\mathrm{prox}} = \rho^Y_k$, and $\rho^Y_k \mathbf{P}^-_{\mathrm{prox}} = \rho^X_{k+1}$.

As mentioned in Section 1.3, we know that $\nu^X$ is stationary for the Proximal Sampler, and from (13), our goal is to then bound the contraction coefficient $\varepsilon_{\mathsf{D}_\Phi}(\mathbf{P}_{\mathrm{prox}}, \nu^X)$. The following lemma shows that this can be bounded by the product of the contraction coefficients of the forward and backward steps.

**Lemma 14** *Let $\mathbf{P}_{\mathrm{prox}} = \mathbf{P}^+_{\mathrm{prox}}\mathbf{P}^-_{\mathrm{prox}}$ denote the Proximal Sampler (9) with joint stationary distribution $\nu^{XY}$. Then,*

$$\varepsilon_{\mathsf{D}_\Phi}(\mathbf{P}_{\mathrm{prox}}, \nu^X) \leq \varepsilon_{\mathsf{D}_\Phi}(\mathbf{P}^+_{\mathrm{prox}}, \nu^X)\,\varepsilon_{\mathsf{D}_\Phi}(\mathbf{P}^-_{\mathrm{prox}}, \nu^Y)\,.$$

**Proof** By Definition 10,

$$
\begin{aligned}
\varepsilon_{\mathsf{D}_\Phi}(\mathbf{P}_{\mathrm{prox}}, \nu^X) &= \sup_\mu \frac{\mathsf{D}_\Phi(\mu\mathbf{P}_{\mathrm{prox}} \,\|\, \nu^X\mathbf{P}_{\mathrm{prox}})}{\mathsf{D}_\Phi(\mu \,\|\, \nu^X)} \\
&= \sup_\mu \left[\frac{\mathsf{D}_\Phi(\mu\mathbf{P}^+_{\mathrm{prox}} \,\|\, \nu^X\mathbf{P}^+_{\mathrm{prox}})}{\mathsf{D}_\Phi(\mu \,\|\, \nu^X)} \times \frac{\mathsf{D}_\Phi(\mu\mathbf{P}^+_{\mathrm{prox}}\mathbf{P}^-_{\mathrm{prox}} \,\|\, \nu^X\mathbf{P}^+_{\mathrm{prox}}\mathbf{P}^-_{\mathrm{prox}})}{\mathsf{D}_\Phi(\mu\mathbf{P}^+_{\mathrm{prox}} \,\|\, \nu^X\mathbf{P}^+_{\mathrm{prox}})}\right] \\
&\leq \varepsilon_{\mathsf{D}_\Phi}(\mathbf{P}^+_{\mathrm{prox}}, \nu^X)\,\varepsilon_{\mathsf{D}_\Phi}(\mathbf{P}^-_{\mathrm{prox}}, \nu^X\mathbf{P}^+_{\mathrm{prox}})\,.
\end{aligned}
$$

The claim follows by noting that $\nu^Y = \nu^X\mathbf{P}^+_{\mathrm{prox}}$ (which is further made clear in Section 4.1). ∎

In light of Lemma 14, we will bound each of the contraction coefficients separately and get our convergence result. As each contraction coefficient can at most equal 1 (as a consequence of data processing inequality), this perspective implies that showing that either coefficient is strictly less than 1 yields a convergence guarantee in Φ-divergence. We will in fact show both coefficients for the forward step and the backward step are strictly less than 1.

## 4.1. Forward step

Without loss of generality consider $k = 0$, and recall the forward step of the Proximal Sampler (9), where $X_0 \sim \rho_0^X$ and $Y_0 \mid X_0 \sim \mathcal{N}(X_0, \eta I)$. To understand the forward step, we wish to understand $\rho_0^Y$. For any $y \in \mathbb{R}^d$, $\rho_0^Y(y) = \int \nu^{Y|X}(y \mid x)\rho_0^X(x)\, \mathrm{d}x = \int \frac{1}{(2\pi\eta)^{\frac{d}{2}}} \exp(-\frac{\|y-x\|^2}{2\eta})\rho_0^X(x)\, \mathrm{d}x$, and therefore $\rho_0^Y = \rho_0^X * \mathcal{N}(0, \eta I)$. This implies that $\mathbf{P}_{\mathrm{prox}}^+$ corresponds to evolving along the heat flow (i.e. $\mathrm{d}X_t = \mathrm{d}W_t$) for time $\eta$. And so, to get a control on $\varepsilon_{\mathsf{D}_\Phi}(\mathbf{P}_{\mathrm{prox}}^+, \nu^X)$, we can use Lemma 8 with $b_t = 0$ and $c = \frac{1}{2}$, just like was done for ULA (discussed in Section 3). This leads to the following guarantee on the contraction coefficient. We prove Lemma 15 in Appendix F.1.

**Lemma 15** *Let* $\mathbf{P}_{\mathrm{prox}} = \mathbf{P}_{\mathrm{prox}}^+ \mathbf{P}_{\mathrm{prox}}^-$ *denote the Proximal Sampler* (9) *with step size* $\eta > 0$, *to sample from* $\nu^X$ *where* $\nu^X$ *satisfies a* $\Phi$-*Sobolev inequality with optimal constant* $\alpha > 0$. *Then,*

$$\varepsilon_{\mathsf{D}_\Phi}(\mathbf{P}_{\mathrm{prox}}^+, \nu^X) \leq \frac{1}{1 + \alpha\eta}.$$

## 4.2. Backward step

We now focus on the backward step of the Proximal Sampler. For the forward step, we were able to relate it to the Brownian motion SDE ($\mathrm{d}X_t = \mathrm{d}W_t$) by observing how it acts on distributions (i.e. on $\rho_0^X$, and seeing that $\rho_0^Y = \rho_0^X * \mathcal{N}(0, \eta I)$). For the backward step, this approach is not as clear. Indeed, writing $\rho_1^X(x) = \int \nu^{X|Y}(x \mid y)\rho_0^Y(y)\, \mathrm{d}y$ does not immediately yield an SDE interpretation. By step 2 of the Proximal Sampler (9), we want to find an SDE such that when initialized from a point mass $\delta_y$ at any $y \in \mathbb{R}^d$, the output has distribution $\nu^{X|Y=y}$ at time $\eta$; and therefore in general, when initialized at $\nu^Y$ the SDE will output $\nu^X$, and when initialized at $\rho_0^Y$ the SDE will output $\rho_1^X$. It turns out we can obtain this by reversing the heat flow path from $\nu^X$ to $\nu^Y$.

For the forward step, we have that $\mathrm{d}X_t = \mathrm{d}W_t$ where if $X_0 \sim \nu^X$, then $X_\eta \sim \nu^Y$. The time reversal of this SDE is called the *backward heat flow* and is given by

$$\mathrm{d}Y_t = \nabla \log(\nu^X * \mathcal{N}_{\eta-t})(Y_t)\, \mathrm{d}t + \mathrm{d}W_t. \tag{15}$$

By construction, if we start (15) from $\nu^Y$, then for any $t \in [0, \eta]$, the marginal law along (15) is the same as $\nu^X * \mathcal{N}_{\eta-t}$. Such a reverse SDE construction is popular in diffusion models (Chen et al., 2023), and details regarding it can be found in Föllmer (2005); Cattiaux et al. (2023). Rigorous connections between (15) and the Proximal Sampler can be found in Chen et al. (2022, Appendix A.1.2); see also the exposition in Chewi (2024, Chapter 8.3) and Kook et al. (2024, Appendix B.2).

The following lemma describes the contraction coefficient for the backward step. We provide the proof of Lemma 16 in Appendix F.2.

**Lemma 16** *Let* $\mathbf{P}_{\mathrm{prox}} = \mathbf{P}_{\mathrm{prox}}^+ \mathbf{P}_{\mathrm{prox}}^-$ *denote the Proximal Sampler* (9) *with step size* $\eta > 0$, *to sample from* $\nu^X$ *where* $\nu^X$ *satisfies a* $\Phi$-*Sobolev inequality with optimal constant* $\alpha > 0$. *Then,*

$$\varepsilon_{\mathsf{D}_\Phi}(\mathbf{P}_{\mathrm{prox}}^-, \nu^Y) \leq \frac{1}{1 + \alpha\eta}.$$

The reason that both Lemmas 15 and 16 have the same result is for two reasons. First, it is because Lemma 8 does not depend on the drift $b_t$. Second, it is because the backward heat flow is the

time reversal of the forward heat flow SDE, and therefore by construction, there is a correspondence between the marginal distributions of the forward and backward processes in the sense that if we start (15) from $\nu^Y$, then for any $t \in [0, \eta]$, the marginal law along (15) is the same as $\nu^X * \mathcal{N}_{\eta-t}$.

### 4.3. Proof of Theorem 3

Equipped with Lemmas 15 and 16, the proof of Theorem 3 follows easily.

**Proof of Theorem 3** By repeated application of Definition 11 and by the fact that $\nu^X$ is stationary for the Proximal Sampler, we have that,

$$\mathsf{D}_\Phi(\rho_k^X \parallel \nu^X) \leq \varepsilon_{\mathsf{D}_\Phi}(\mathbf{P}_{\mathrm{prox}}, \nu^X)^k \, \mathsf{D}_\Phi(\rho_0^X \parallel \nu^X).$$

Further using Lemmas 14, 15, and 16, we get,

$$\mathsf{D}_\Phi(\rho_k^X \parallel \nu^X) \leq \frac{\mathsf{D}_\Phi(\rho_0^X \parallel \nu^X)}{(1 + \alpha\eta)^{2k}}.$$

$\blacksquare$

An explicit calculation for the Ornstein-Uhlenbeck process implying the tightness of Theorem 3 for KL divergence can be found in Proposition 18 in Appendix G.2.

## 5. Discussion

We study the mixing time in $\Phi$-divergence for two popular discretizations of the Langevin dynamics, namely the Unadjusted Langevin Algorithm (ULA) and the Proximal Sampler. Our results show mixing to the stationary distributions of these Markov chains under the stationary distributions satisfying $\Phi$-Sobolev inequalities. As the Proximal Sampler is unbiased, this implies mixing to the target distribution $\nu$ of interest. However, this is not the case for the ULA, where our mixing guarantees are to the biased limit $\nu^\eta$ of ULA. While $\nu^\eta$ can be shown to satisfy a $\Phi$-Sobolev inequality under strong log-concavity of $\nu$ (Lemma 13), it is interesting to study what can hold under weaker assumptions on $\nu$. For example one can ask if the fact that $\nu^\eta$ satisfies a $\Phi$-Sobolev inequality is closed under certain perturbations of $\nu$. Alternatively, one can also ask for biased convergence guarantees of ULA to $\nu$ in $\Phi$-divergence directly. One can also pose all of these questions for other Markov chains, such as Hamiltonian Monte Carlo and the underdamped Langevin algorithm.

Our results are based on strong data processing inequalities in $\Phi$-divergence, and proceed by bounding the respective contraction coefficients. However, different forms of (strong) data processing inequalities exist, namely in terms of mutual information and $\Phi$-mutual information. It is interesting to ask if our approaches can extend to these other forms of strong data processing inequalities, to yield for example, the convergence of the mutual information functional. Additionally, we require the function $\Phi$ to be twice differentiable, which prohibits many popular $\Phi$-divergences such as total variation distance and "hockey-stick" divergences (Sason and Verdú, 2016). These divergences are popular in differential privacy applications (Asoodeh et al., 2020) and it would therefore be instructive to study extensions to such divergences directly.

Finally, our results imply mixing in all (twice-differentiable) $\Phi$-divergences for strongly log-concave target distributions. It is interesting to ask if mixing bounds can be obtained in a variety of $\Phi$-divergences under weaker absolute isoperimetric assumptions on $\nu$ (such as a log-Sobolev inequality, or in other words, a $\Phi$-Sobolev inequality with $\Phi(x) = x \log x$).

## Acknowledgments

We thank Varun Jog, Sinho Chewi, and Yuliang Wang for helpful comments and references. S.M. and A.W. were supported by NSF Award CCF #2403391.

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

## Appendix A. Examples of $\Phi$-divergences

| $\Phi(x)$ | $\mathbf{D_\Phi(\mu \parallel \nu)}$ | $\mathbf{D_\Phi(\mu \parallel \nu)}$ name | $\Phi$-Sobolev inequality |
|---|---|---|---|
| $x \log x$ | $\int \mathrm{d}\mu \log \frac{\mathrm{d}\mu}{\mathrm{d}\nu}$ | KL divergence | log-Sobolev inequality |
| $(x-1)^2$ | $\int \frac{(\mathrm{d}\mu - \mathrm{d}\nu)^2}{\mathrm{d}\nu}$ | chi-squared divergence | Poincaré inequality |
| $\frac{1}{2}(\sqrt{x}-1)^2$ | $\frac{1}{2}\int(\sqrt{\mathrm{d}\mu} - \sqrt{\mathrm{d}\nu})^2$ | squared Hellinger distance | – |
| $\frac{1}{2}|x-1|$ | $\frac{1}{2}\int |\mathrm{d}\mu - \mathrm{d}\nu|$ | TV distance | – |
| $-\log x$ | $\int \mathrm{d}\nu \log \frac{\mathrm{d}\nu}{\mathrm{d}\mu}$ | reverse KL divergence | – |
| $\frac{1}{x} - x$ | $2 + \int \frac{(\mathrm{d}\mu - \mathrm{d}\nu)^2}{\mathrm{d}\mu}$ | reverse chi-squared divergence | – |

Table 1: Common $\Phi$ functions along with corresponding $\Phi$-divergences (1) and $\Phi$-Sobolev inequalities (Definition 1).

## Appendix B. Restricted Gaussian Oracle

Here we show a basic implementation of the RGO via rejection sampling. As mentioned in Section 1.3, enhanced implementations of the RGO under weaker assumptions have been an active area of research and we refer the reader to Fan et al. (2023) and the references therein.

Recall from Section 1.3 the conditional distribution the RGO seeks to sample from:

$$\nu^{X|Y}(x \mid y) \propto_x \exp\left(-f(x) - \frac{\|x-y\|^2}{2\eta}\right).$$

Define $g_y(x) \coloneqq f(x) + \frac{\|x-y\|^2}{2\eta}$ so that for any fixed $y \in \mathbb{R}^d$, the target distribution for the RGO is $\tilde{\nu}_y(x) \propto \exp(-g_y(x))$. Suppose the potential function $f$ is $L$-smooth and that $\eta < \frac{1}{L}$. In this case, $\tilde{\nu}_y$ is strongly log-concave with condition number $\frac{1+L\eta}{1-L\eta}$. and the RGO can be implemented efficiently via rejection sampling.

Suppose $\pi \propto \exp(-V)$ is $\beta$-strongly log-concave and $M$-smooth. The rejection sampling method to sample from $\pi$ is the following:

1. Compute the minimizer $x^*$ of $V$, so that for any $z \in \mathbb{R}^d$, $V(z) \geq V(x^*) + \frac{\beta}{2}\|z - x^*\|^2$.

2. Draw $Z \sim \mathcal{N}(x^*, \frac{1}{\beta}I)$ and accept it with probability

$$\exp\left(-V(Z) + V(x^*) + \frac{\beta}{2}\|Z - x^*\|^2\right).$$

Repeat this until acceptance.

The output of this method is distributed according to $\pi$ and the expected number of iterations is $(\frac{M}{\beta})^{d/2}$ (Chewi et al., 2022b, Theorem 7).

Applying this to sample from $\tilde{\nu}_y$ with $\eta \asymp \frac{1}{Ld}$ gives a valid implementation of the RGO under smoothness of $f$ with $\mathcal{O}(1)$ many iterations in expectation. Specifically, $M = L + \frac{1}{\eta}$ and $\beta = -L + \frac{1}{\eta}$ and therefore, $\frac{M}{\beta} = \frac{1+L\eta}{1-L\eta}$. So if $\eta = \frac{1}{Ld}$, $(\frac{M}{\beta})^{d/2} = (1 + \frac{2}{d-1})^{d/2} = \mathcal{O}(1)$.

## Appendix C. Proof of Lemma 8

**Proof** Begin by recalling that if $X_t \sim \rho_t$ where $\mathrm{d}X_t = b_t(X_t)\,\mathrm{d}t + \sqrt{2c}\,\mathrm{d}W_t$, then $\rho_t : \mathbb{R}^d \to \mathbb{R}$ satisfies the Fokker-Planck equation, given by:

$$\partial_t \rho_t = -\nabla \cdot (\rho_t\, b_t) + c\Delta\rho_t \,.$$

Also note that using the identity $\Delta\rho = \nabla \cdot (\rho\nabla\log\rho)$, the above can be written as:

$$\partial_t \rho_t = -\nabla \cdot (\rho_t\, b_t) + c\nabla \cdot (\rho_t\nabla\log\rho_t) \,. \tag{16}$$

We identify $\mu_t$ and $\nu_t$ with their densities with respect to Lebesgue measure, and further denote their relative density as $h_t = \frac{\mu_t}{\nu_t}$. We also assume enough regularity to take the differential under the integral sign and use $\int fg$ as a shorthand for $\int f(x)g(x)\,\mathrm{d}x$. Throughout the proof, we use integration by parts in various steps, denoted by (IBP). With all of this in mind, we have the following:

$$\partial_t \, \mathsf{D}_\Phi(\mu_t \,\|\, \nu_t) = \partial_t \int \nu_t \, \Phi(h_t)$$

$$= \int (\partial_t \nu_t)\Phi(h_t) + \int \nu_t \left(\partial_t \, \Phi(h_t)\right)$$

$$= \int (\partial_t \nu_t)\Phi(h_t) + \int \nu_t \Phi'(h_t)\frac{\nu_t\,\partial_t\mu_t - \mu_t\,\partial_t\nu_t}{\nu_t^2}$$

$$= \underbrace{\int (\partial_t \nu_t)\Phi(h_t)}_{T_1} + \underbrace{\int \Phi'(h_t)(\partial_t\mu_t)}_{T_2} - \underbrace{\int \Phi'(h_t)\frac{\mu_t}{\nu_t}(\partial_t\nu_t)}_{T_3}$$

We will now handle each of these terms separately. We have:

$$T_1 = \int (\partial_t \nu_t)\Phi(h_t)$$

$$\stackrel{(16)}{=} \int \left(-\nabla \cdot (\nu_t\, b_t) + c\nabla \cdot (\nu_t\nabla\log\nu_t)\right)\Phi(h_t)$$

$$= -\int \nabla \cdot (\nu_t\, b_t)\Phi(h_t) + c\int \nabla \cdot (\nu_t\nabla\log\nu_t)\Phi(h_t)$$

$$\stackrel{(\mathrm{IBP})}{=} \int \langle \nu_t b_t, \nabla(\Phi(h_t))\rangle - c\int \langle \nu_t\nabla\log\nu_t, \nabla(\Phi(h_t))\rangle$$

$$= \int \langle \nu_t b_t, \Phi'(h_t)\nabla\frac{\mu_t}{\nu_t}\rangle - c\int \langle \nu_t\nabla\log\nu_t, \Phi'(h_t)\nabla\frac{\mu_t}{\nu_t}\rangle$$

We also have:

$$T_2 = \int \Phi'(h_t)(\partial_t \mu_t)$$

$$\stackrel{(16)}{=} \int \Phi'(h_t)\left(-\nabla \cdot (\mu_t\, b_t) + c\nabla \cdot (\mu_t\nabla \log \mu_t)\right)$$

$$\stackrel{\text{(IBP)}}{=} \int \langle \mu_t\, b_t, \nabla(\Phi'(h_t)) \rangle - c \int \langle \mu_t\nabla \log \mu_t, \nabla(\Phi'(h_t)) \rangle$$

$$= \int \langle \mu_t\, b_t, \Phi''(h_t)\nabla \frac{\mu_t}{\nu_t} \rangle - c \int \langle \mu_t\nabla \log \mu_t, \Phi''(h_t)\nabla \frac{\mu_t}{\nu_t} \rangle$$

We also have:

$$T_3 = \int \Phi'(h_t)\frac{\mu_t}{\nu_t}(\partial_t \nu_t)$$

$$\stackrel{(16)}{=} \int \Phi'(h_t)\frac{\mu_t}{\nu_t}\left(-\nabla \cdot (\nu_t\, b_t) + c\nabla \cdot (\nu_t\nabla \log \nu_t)\right)$$

$$\stackrel{\text{(IBP)}}{=} \int \left\langle \nu_t\, b_t, \nabla\left(\Phi'(h_t)\frac{\mu_t}{\nu_t}\right) \right\rangle - c \int \left\langle \nu_t\nabla \log \nu_t, \nabla\left(\Phi'(h_t)\frac{\mu_t}{\nu_t}\right) \right\rangle$$

$$= \int \langle \mu_t\, b_t, \Phi''(h_t)\nabla \frac{\mu_t}{\nu_t} \rangle + \int \langle \nu_t b_t, \Phi'(h_t)\nabla \frac{\mu_t}{\nu_t} \rangle$$

$$- c \int \left\langle \mu_t\nabla \log \nu_t, \Phi''(h_t)\nabla \frac{\mu_t}{\nu_t} \right\rangle - c \int \left\langle \nu_t\nabla \log \nu_t, \Phi'(h_t)\nabla \frac{\mu_t}{\nu_t} \right\rangle$$

Therefore, combining the above, we see many terms cancel and we have the following:

$$\partial_t\, \mathsf{D}_\Phi(\mu_t \parallel \nu_t) = T_1 + T_2 - T_3$$

$$= c \int \left\langle \mu_t\nabla \log \nu_t, \Phi''(h_t)\nabla \frac{\mu_t}{\nu_t} \right\rangle - c \int \left\langle \mu_t\nabla \log \mu_t, \Phi''(h_t)\nabla \frac{\mu_t}{\nu_t} \right\rangle$$

$$= -c \int \left\langle \nabla \log \frac{\mu_t}{\nu_t}, \Phi''(h_t)\nabla \frac{\mu_t}{\nu_t} \right\rangle \mu_t$$

$$= -c\, \mathop{\mathbb{E}}_{\mu_t}\left[ \left\langle \nabla \log \frac{\mu_t}{\nu_t}, \Phi''(h_t)\nabla \frac{\mu_t}{\nu_t} \right\rangle \right]$$

$$= -c\, \mathop{\mathbb{E}}_{\nu_t}\left[ \frac{\mu_t}{\nu_t} \left\langle \nabla \log \frac{\mu_t}{\nu_t}, \Phi''(h_t)\nabla \frac{\mu_t}{\nu_t} \right\rangle \right]$$

$$= -c\, \mathop{\mathbb{E}}_{\nu_t}\left[ \left\langle \nabla \frac{\mu_t}{\nu_t}, \Phi''(h_t)\nabla \frac{\mu_t}{\nu_t} \right\rangle \right]$$

$$= -c\, \mathsf{FI}_\Phi(\mu_t \parallel \nu_t)\,,$$

which proves the desired statement. ∎

## Appendix D. Proof Of Lemma 12

**Proof** Let $\mu$ be an arbitrary probability distribution such that $\mathsf{D}_\Phi(\mu \parallel \nu^\eta) < \infty$. As $\nu$ is $L$-smooth (i.e. $-LI \preceq \nabla^2 f \preceq LI$) and $\eta \leq \frac{1}{L}$, $F(x) = x - \eta\nabla f(x)$ is a bijective map. Furthermore, it also holds that $\|F\|_{\mathsf{Lip}} \leq 1 + \eta L$.

Therefore, from Lemma 5, we see that $\alpha_{\Phi\mathsf{SI}}(F_\#\nu^\eta) \geq \frac{\alpha}{(1+\eta L)^2}$. Now for $t \geq 0$, let $\mu_t := F_\#\mu * \mathcal{N}(0, tI)$ and $\nu_t := F_\#\nu^\eta * \mathcal{N}(0, tI)$. For $t \geq 0$, further denote $\alpha_{\Phi\mathsf{SI}}(\nu_t) = \alpha_t$ as shorthand. Also note that $\mathcal{N}(0, tI)$ is $1/t$-strongly log-concave and therefore from Lemma 7, $\alpha_{\Phi\mathsf{SI}}(\mathcal{N}(0, tI)) \geq 1/t$. Hence, Lemma 6 implies that

$$\alpha_t \geq \frac{\alpha}{(1+\eta L)^2 + \alpha t}\,.$$

The rate of change of Φ-divergence between $\mu_t$ and $\nu_t$ is given by Lemma 8. Applying Lemma 8 for (10) with $b_t = 0$ and $c = \frac{1}{2}$, which is what the (Gaussian convolution) evolution of $\mu_t$ and $\nu_t$ corresponds to, implies,

$$\frac{\mathrm{d}}{\mathrm{d}t}\mathsf{D}_\Phi(\mu_t \parallel \nu_t) = -\frac{1}{2}\mathsf{FI}_\Phi(\mu_t \parallel \nu_t)$$
$$\leq -\alpha_t \mathsf{D}_\Phi(\mu_t \parallel \nu_t),$$

where the inequality follows from $\nu_t$ satisfying a Φ-Sobolev inequality. Applying Grönwall's lemma and integrating the differential inequality from $0$ to $2\eta$ yields,

$$\frac{\mathsf{D}_\Phi(\mu_{2\eta} \parallel \nu_{2\eta})}{\mathsf{D}_\Phi(\mu_0 \parallel \nu_0)} \leq \exp\left(-\int_0^{2\eta} \alpha_t\, \mathrm{d}t\right) \leq \frac{(1+\eta L)^2}{(1+\eta L)^2 + 2\alpha\eta}\,.$$

Finally, note that $\mathsf{D}_\Phi(\mu_{2\eta} \parallel \nu_{2\eta}) = \mathsf{D}_\Phi(\mu\mathbf{P} \parallel \nu^\eta\mathbf{P})$ and $\mathsf{D}_\Phi(\mu_0 \parallel \nu_0) = \mathsf{D}_\Phi(\mu \parallel \nu^\eta)$, where the latter holds as $F$ is bijective. This, along with the fact that $\mu$ was arbitrary completes the proof. ∎

## Appendix E. Proof of Lemma 13

**Proof** Recall the ULA update in law (14). Under the assumptions of $\alpha$-strong log-concavity, $L$-smoothness, and $\eta \leq \frac{1}{L}$, it holds that $F(x) = x - \eta\nabla f(x)$ is a bijective map with $\|F\|_{\mathsf{Lip}} \leq 1 - \alpha\eta$. Also note that $\mathcal{N}_{2\eta}$ is $\frac{1}{2\eta}$-SLC and therefore satisfies a Φ-Sobolev inequality with the same constant (Lemma 7). Suppose $\rho_k$ satisfies $\alpha_k$-ΦSI. Then Lemmas 5 and 6 imply that

$$\frac{1}{\alpha_{k+1}} \leq \frac{(1-\alpha\eta)^2}{\alpha_k} + 2\eta\,.$$

Therefore, suppose we start from $\rho_0$ such that $\alpha_0 \geq \frac{\alpha}{2}$. Then by induction $\alpha_k \geq \frac{\alpha}{2}$ for all $k \geq 0$. Indeed, let us show that if $\alpha_k \geq \frac{\alpha}{2}$, then $\alpha_{k+1} \geq \frac{\alpha}{2}$. This follows as:

$$\frac{1}{\alpha_{k+1}} \leq \frac{2(1-\alpha\eta)^2}{\alpha} + 2\eta = \frac{2}{\alpha}(1 - \alpha\eta(1-\alpha\eta)) \leq \frac{2}{\alpha}\,.$$

Therefore, taking $k \to \infty$, we get that for the limiting distribution, $\alpha_{\Phi\mathsf{SI}}(\nu^\eta) \geq \frac{\alpha}{2}$. ∎

## Appendix F. Deferred Proofs From Section 4

### F.1. Proof of Lemma 15

**Proof** Let $\mu$ be an arbitrary distribution such that $\mathsf{D}_\Phi(\mu \parallel \nu^X) < \infty$. Define $\mu_t := \mu * \mathcal{N}_t$ and $\nu_t := \nu^X * \mathcal{N}_t$. Further denote $\alpha_t$ as shorthand for $\alpha_{\Phi\mathsf{SI}}(\nu_t)$. By assumption, we know that $\nu^X$

satisfies a $\Phi$-Sobolev inequality with optimal constant $\alpha$. Also note that $\mathcal{N}_t$ is $\frac{1}{t}$-SLC and therefore satisfies a $\Phi$-Sobolev inequality with the same constant (Lemma 7). Therefore by Lemma 6 we have that

$$\alpha_t \geq \frac{\alpha}{1 + \alpha t}. \tag{17}$$

Hence, applying Lemma 8 on the SDE $\mathrm{d}X_t = \mathrm{d}W_t$ gives us,

$$\frac{\mathrm{d}}{\mathrm{d}t} \mathsf{D}_\Phi(\mu_t \,\|\, \nu_t) = -\frac{1}{2}\mathsf{FI}_\Phi(\mu_t \,\|\, \nu_t).$$

We can then apply the $\Phi$-Sobolev inequality of $\nu_t$ and integrate the differential inequality from $0$ to $\eta$ to yield

$$\frac{\mathsf{D}_\Phi(\mu_\eta \,\|\, \nu_\eta)}{\mathsf{D}_\Phi(\mu_0 \,\|\, \nu_0)} \leq \exp\left(-\int_0^\eta \alpha_t\,\mathrm{d}t\right) \leq \frac{1}{1 + \alpha\eta}.$$

Observe that $\mathsf{D}_\Phi(\mu_\eta \,\|\, \nu_\eta) = \mathsf{D}_\Phi(\mu\mathbf{P}^+ \,\|\, \nu^X\mathbf{P}^+)$ and that $\mathsf{D}_\Phi(\mu_0 \,\|\, \nu_0) = \mathsf{D}_\Phi(\mu \,\|\, \nu^X)$. As $\mu$ was arbitrary, this gives a valid bound on the contraction coefficient and concludes the proof. $\blacksquare$

### F.2. Proof of Lemma 16

**Proof** The proof follows similarly to that of Lemma 15. Let $\mu$ be an arbitrary distribution such that $\mathsf{D}_\Phi(\mu \,\|\, \nu^Y) < \infty$. Define $\mu_t$ to be the marginal law at time $t$ when starting the SDE (15) from $\mu$, and let $\nu_t$ be the marginal law at time $t$ when starting the SDE (15) from $\nu^Y$. Further denote $\alpha_t$ as shorthand for $\alpha_{\Phi\mathsf{SI}}(\nu_t)$. As by construction of the SDE (15), $\nu_t = \nu^X * \mathcal{N}_{\eta-t}$, the same argument used to derive (17) yields that

$$\alpha_t \geq \frac{\alpha}{1 + \alpha(\eta - t)}.$$

Hence, applying Lemma 8 on the SDE (15) gives us,

$$\frac{\mathrm{d}}{\mathrm{d}t} \mathsf{D}_\Phi(\mu_t \,\|\, \nu_t) = -\frac{1}{2}\mathsf{FI}_\Phi(\mu_t \,\|\, \nu_t).$$

We can then apply the $\Phi$-Sobolev inequality of $\nu_t$ and integrate the differential inequality from $0$ to $\eta$ to yield

$$\frac{\mathsf{D}_\Phi(\mu_\eta \,\|\, \nu_\eta)}{\mathsf{D}_\Phi(\mu_0 \,\|\, \nu_0)} \leq \exp\left(-\int_0^\eta \alpha_t\,\mathrm{d}t\right) \leq \frac{1}{1 + \alpha\eta}.$$

Observe that $\mathsf{D}_\Phi(\mu_\eta \,\|\, \nu_\eta) = \mathsf{D}_\Phi(\mu\mathbf{P}^- \,\|\, \nu^Y\mathbf{P}^-)$ and that $\mathsf{D}_\Phi(\mu_0 \,\|\, \nu_0) = \mathsf{D}_\Phi(\mu \,\|\, \nu^Y)$. As $\mu$ was arbitrary, this gives a valid bound on the contraction coefficient and concludes the proof. $\blacksquare$

## Appendix G. Ornstein-Uhlenbeck Process

Recall the Langevin dynamics (5) to sample from $\nu \propto \exp(-f)$. When $\nu$ is a Gaussian, the Langevin dynamics is known as the Ornstein-Uhlenbeck process. For example, when $\nu = \mathcal{N}(0, \frac{1}{\alpha}I)$, the Langevin dynamics (5) becomes:

$$\mathrm{d}X_t = -\alpha X_t\,\mathrm{d}t + \sqrt{2}\,\mathrm{d}W_t.$$

When the initial distribution $\rho_0$ is a Gaussian, the marginal distributions $\rho_t$ admit convenient Gaussian forms. We will leverage this to show that Theorems 2 and 3 are tight for KL divergence.

Throughout, let $\nu = \mathcal{N}(0, \frac{1}{\alpha}I)$ for some $\alpha > 0$. Recall that this is $\alpha$-strongly log-concave, and hence also satisfies a $\Phi$-Sobolev inequality with the same constant (Lemma 7).

### G.1. ULA evolution

The ULA update (6) for $\nu = \mathcal{N}(0, \frac{1}{\alpha}I)$ takes the following form:

$$X_{k+1} = (1 - \alpha\eta)X_k + \sqrt{2\eta}Z_k, \tag{18}$$

and the solution to (18) is:

$$X_k = (1 - \alpha\eta)^k X_0 + \sqrt{\frac{2(1 - (1-\alpha\eta)^{2k})}{\alpha(2 - \alpha\eta)}}Z, \tag{19}$$

where $Z \sim \mathcal{N}(0, I)$. In this case, the biased limit $\nu^\eta$ is:

$$\nu^\eta = \mathcal{N}\left(0, \frac{2}{\alpha(2 - \alpha\eta)}I\right). \tag{20}$$

To show a tightness result, we simply take $\rho_0 = \mathcal{N}(\mathbf{1}, I)$ where $\mathbf{1}$ is the $d$-dimensional all-ones vector.

**Proposition 17** *Consider the ULA* (6) *for* $\nu = \mathcal{N}(0, \frac{1}{\alpha}I)$ *with* $\rho_0 = \mathcal{N}(\mathbf{1}, I)$ *and* $\eta \leq \frac{1}{\alpha}$. *Then,* $\mathsf{KL}(\rho_k \parallel \nu^\eta) = \mathcal{O}(d\alpha(1 - \eta\alpha)^{2k})$.

**Proof** From (19), we can see that $X_k \sim \rho_k$ where

$$\rho_k = \mathcal{N}\left((1 - \eta\alpha)^k \mathbf{1}, \frac{2 + (1 - \eta\alpha)^{2k}(2\alpha - \eta\alpha^2 - 2)}{\alpha(2 - \eta\alpha)}I\right). \tag{21}$$

Taking $k \to \infty$ in (21), we see that the biased limit is indeed given by (20), i.e.,

$$\nu^\eta = \mathcal{N}\left(0, \frac{2}{\alpha(2 - \alpha\eta)}I\right).$$

Therefore, using the formula for KL divergence between two multivariate Gaussians, we get

$$\mathsf{KL}(\rho_k\|\nu^\eta) = \frac{d}{2}\left[\frac{(1 - \eta\alpha)^{2k}\alpha(2 - \eta\alpha)}{2} + \frac{(1 - \eta\alpha)^{2k}(2\alpha - \eta\alpha^2 - 2)}{2} - \log\left(1 + \frac{(1 - \eta\alpha)^{2k}(2\alpha - \eta\alpha^2 - 2)}{2}\right)\right].$$

This simplifies to reveal that

$$\mathsf{KL}(\rho_k \parallel \nu^\eta) = \mathcal{O}(d\alpha(1 - \eta\alpha)^{2k}).$$

■

### G.2. Proximal Sampler evolution

Similar to the ULA setting in Appendix G.1, we will take the target distribution to be $\nu^X = \mathcal{N}(0, \frac{1}{\alpha}I)$ and the starting distribution to be $\rho_0^X = \mathcal{N}(\mathbf{1}, I)$ where $\mathbf{1}$ is the $d$-dimensional all-ones vector.

The Proximal Sampler update (9) reveals that $\rho_0^Y = \mathcal{N}(\mathbf{1}, (1+\eta)I)$. Moreover, we also have that for any $y \in \mathbb{R}^d$,

$$\nu^{X|Y}(\cdot \mid y) = \mathcal{N}\left(\frac{y}{1+\eta\alpha}, \frac{\eta}{1+\eta\alpha}I\right),$$

and consequently, as $\rho_1^X(x) = \int \nu^{X|Y}(x \mid y)\rho_0^Y(y)\,\mathrm{d}y$, $\rho_1^X$ is Gaussian. This reasoning extends to show that all of the iterates $\{\rho_k^X\}$ and $\{\rho_k^Y\}$ for $k \geq 0$ are Gaussian. Denoting $\rho_k^X = \mathcal{N}(m_k, c_k I)$, where for all $k \geq 0$, $m_k \in \mathbb{R}^d$ and $c_k > 0$, we have that,

$$m_{k+1} = \frac{m_k}{1+\eta\alpha} \quad \text{and} \quad c_{k+1} = \frac{1}{(1+\eta\alpha)^2}\left(c_k - \frac{1}{\alpha}\right) + \frac{1}{\alpha}.$$

Therefore, as $m_0 = \mathbf{1}$ and $c_0 = 1$,

$$m_k = \frac{1}{(1+\eta\alpha)^k}\mathbf{1} \quad \text{and} \quad c_k = \frac{1}{(1+\eta\alpha)^{2k}}\left(1 - \frac{1}{\alpha}\right) + \frac{1}{\alpha}. \tag{22}$$

We have the following proposition.

**Proposition 18** *Consider the Proximal Sampler (9) for $\nu^X = \mathcal{N}(0, \frac{1}{\alpha}I)$ with $\rho_0^X = \mathcal{N}(\mathbf{1}, I)$ and $\eta > 0$. Then, $\mathsf{KL}(\rho_k \parallel \nu^\eta) = \mathcal{O}(d\alpha(1-\eta\alpha)^{-2k})$.*

**Proof** Recall from (22) that $\rho_k^X = \mathcal{N}(m_k, c_k I)$ with

$$m_k = \frac{1}{(1+\eta\alpha)^k}\mathbf{1} \quad \text{and} \quad c_k = \frac{1}{(1+\eta\alpha)^{2k}}\left(1 - \frac{1}{\alpha}\right) + \frac{1}{\alpha}.$$

Therefore, using the formula for KL divergence between two multivariate Gaussians, we get

$$\mathsf{KL}(\rho_k^X \parallel \nu^X) = \frac{d}{2}\left[\frac{\alpha}{(1+\eta\alpha)^{2k}} + \frac{\alpha-1}{(1+\eta\alpha)^{2k}} - \log\left(1 + \frac{\alpha-1}{(1+\eta\alpha)^{2k}}\right)\right].$$

This simplifies to reveal that

$$\mathsf{KL}(\rho_k \parallel \nu^\eta) = \mathcal{O}(d\alpha(1-\eta\alpha)^{-2k}).$$

∎

**Conclusion** From Corollary 9, Proposition 17, and Proposition 18, we can see that the rates of convergence of KL divergence along the continuous time Langevin dynamics, ULA, and the Proximal Sampler are $(d\alpha\exp(-2\alpha t))$, $(d\alpha(1-\alpha\eta)^{2k})$, and $(d\alpha(1+\alpha\eta)^{-2k})$ respectively.

