# OpenReview forum: "Fast Convergence of $\Phi$-Divergence Along the Unadjusted Langevin Algorithm and Proximal Sampler"
_algorithmiclearningtheory.org/ALT/2025/Conference — ALT 2025_

### Official Review · Reviewer_7k7y · 2024-10-31

**Rating:** 6
**Confidence:** 5

**Review:**

The authors studied the convergence rates of the unadjusted Langevin algorithm (ULA) and the proximal sampler using the measure of $\Phi$-divergences. The class of $\Phi$-divergences cover a wide range of metrics and divergences such as KL, chi-squared, TV, and Hellinger.

The generality and the tightness of both of the main convergence results are very welcomed. This is a nice unification of many different analyses arising from different papers, but conveniently and cleanly summarized into these general results. While the proofs are not the necessarily the most complex, both main results are fairly clean extensions of Vempala and Wibisono (2019) and Chen et al. (2022). I did not carefully analyze every step of the proof, but the framework makes sense to me, and the results are strongly believable as they recover existing results for KL divergence.

I have several questions for the authors.

1. The convergence result of Theorem 1 for ULA is for the biased stationary distribution of ULA $\nu^\eta$. While this result is interesting in its own right, I believe the most desirable type of guarantee is for $\nu$. Essentially, there are two components missing to achieve this: (a) a bias characterizing the "distance" between $\nu$ and $\nu^\eta$ (b) a type of "triangle inequality" for the $\Phi$-divergence to characterize the gap using this bias. Can the authors firstly comment on the technical challenges involved with this extension? Secondly, can the authors add a discussion on this limitation in the main text, in particular with respect to the dimension dependence of the bias? I will not hold this second question negatively against this paper, as I simply prefer the limitations to be clearly communicated.

2. While the $\Phi$-divergences cover a large class of desirable divergences, can the authors discuss the possible extension to the Renyi divergences? There is a sense that Renyi divergence is a better measure of convergence guarantees than chi-squared, in particular the initial conditions will not introduce an exponential dependence on dimension.

3. The main results on proximal sampler assumes access to the RGO, which is why the current convergence guarantee is independent of dimension. Can the authors comment on how some of the numerical procedure to implementing the RGO might introduce additional dimension and other dependence for the convergence guarantee? In particular, would we recover the results of Chen et al. (2022)?

A couple of further minor points.

1. Can the authors move the definition of the $\Phi$-Sobolev inequality up to before the Theorem statements, as it is the most important assumption of this paper?
2. Would the authors require $\Phi$ to be twice differentiable as the Fisher information now contains $\Phi''$? If it is not required, then for the case of total variation for example, how should I interpret the Fisher information with a delta functional?

At this point, I will recommend a "weak accept", but I would be happy to raise the score if the authors can address my questions above.

**Paper Award:**

No

---

> ### Author Response · Authors · 2024-11-23
>
> We thank the reviewer for their review and feedback. We address the specific questions below.
>
> * Showing mixing guarantees via SDPI crucially requires the stationary distribution to be the second argument in the $\Phi$-divergence (cf. (13) in the paper). We agree it would be interesting to extend it to a biased convergence result to $\nu$, but currently it seems challenging. We would like to note that the mixing of ULA to the biased limit $\nu^\eta$ has been the subject of prior works (including [Vempala, Wibisono 23] and [Altschuler, Talwar 23]).\
> The dependence of dimension $d$ in the bias will greatly depend on the function $\Phi$ that we choose. For example, consider ULA with Gaussian target distribution $\nu = \mathcal{N}(0, \frac{1}{\alpha}I)$, which is $\alpha$-strongly log-concave. The biased limit of ULA in this case is $\nu^{\eta} = \mathcal{N}\big(0, \frac{2}{\alpha(2-\eta \alpha)}I\big)$.
> Therefore, we can see that (omitting dependence on other variables), $\mathsf{KL}(\nu^{\eta} || \nu) = O(d)$ whereas $\chi^2(\nu^{\eta} || \nu) = O(\exp(d))$.\
> The triangle inequality for $\Phi$-divergences is an interesting but challenging question, and we are only aware of specific $\Phi$-divergences satisfying it. Apart from the total variation distance, Marton's divergence [(72) in Sason, Verdú 16] is an example of a $\Phi$-divergence satisfying triangle inequality. However, the square root of many $\Phi$-divergences (such as squared-Hellinger divergence, Jenson-Shannon divergence, Le Cam divergence) satisfy the triangle inequality [Chapter 7 in Polyanskiy, Wu 24].\
> We will add a discussion on the above points in our revision.
>
>
> * As Rényi divergences are not $\Phi$-divergences, our results do not directly imply convergence in Rényi divergence. However, the convergence of the ULA and proximal sampler in Rényi divergence under a log-Sobolev inequality or Poincaré inequality assumption on the stationary distribution can be shown via an SDPI based proof (similar to what we show for $\Phi$-divergences). These results for ULA and proximal sampler are covered in [Vempala, Wibisono 23] and [Chen, Chewi, Salim, Wibisono 22] respectively, albeit not explicitly stated in terms of SDPIs and contraction coefficients. We will mention these results in the final version of our paper.
>
>
> * We only briefly mention an RGO implementation via rejection sampling in Appendix B. Assuming further that $\nu^X \propto \exp(-f)$ is $L$-smooth, the rejection sampling approach in Appendix B shows that with $\eta \asymp \frac{1}{Ld}$, each call to the rejection sampling based RGO implementation requires $O(1)$ many oracle calls to $f$. Therefore, we can show the following corollary.\
> **Corollary:** Suppose $\nu^X \propto \exp(-f)$ satisfies a $\Phi$-Sobolev inequality with optimal constant $\alpha$ and is $L$-smooth. Then for any $\epsilon > 0$, the proximal sampler with $\eta \asymp \frac{1}{Ld}$ and with rejection sampling based RGO implementation (as described in Appendix B) outputs $X_k \sim \rho_k^X$ such that $\mathsf{D}_{\Phi} (\rho_k^X || \nu^X) \leq \epsilon $ and the expected number of oracle calls to $f$ is $O\big( \frac{Ld}{\alpha} \log  \frac{ \mathsf{D}{\Phi}  (\rho_0^X || \nu^X) }{\epsilon}  \big)$.\
> This corollary is similar to the results found in Section 4.2 in [Chen, Chewi, Salim, Wibisono 22] and we will add it in the revised draft.
>
> * We will move the definition of $\Phi$-Fisher information and consequently, of $\Phi$-Sobolev inequality before stating the theorems.
>
> * The reviewer is correct, in this paper we require the twice differentiability due to the $\Phi$-Fisher information functional. It might be possible to consider the TV distance as a limit of a smoothed approximation of $\Phi$-divergence; we will investigate this in future work.
>
>
> [Vempala, Wibisono 23] Santosh S. Vempala and Andre Wibisono. Rapid Convergence of the Unadjusted Langevin Algorithm: Isoperimetry Suffices. Geometric Aspects of Functional Analysis, 2023.
>
> [Chen, Chewi, Salim, Wibisono 22] Yongxin Chen, Sinho Chewi, Adil Salim, and Andre Wibisono. Improved analysis for a proximal algorithm for sampling. In Conference on Learning Theory, pages 2984–3014. PMLR, 2022
>
> [Altschuler, Talwar 23] Altschuler, Jason, and Kunal Talwar. Resolving the Mixing Time of the Langevin Algorithm to its Stationary Distribution for Log-Concave Sampling. The Thirty Sixth Annual Conference on Learning Theory. PMLR, 2023.
>
>
> [Sason, Verdú 16] Sason, Igal, and Sergio Verdú. "$ f $-divergence Inequalities." IEEE Transactions on Information Theory 62.11 (2016): 5973-6006.
>
>
> [Polyanskiy, Wu 24] Yury Polyanskiy and Yihong Wu. Information Theory: From Coding to Learning. Cambridge University Press, 2024.

---

> > ### Comment · Reviewer_7k7y · 2024-11-25
> > **Response**
> >
> > Thank you for the detailed response. Please include the discussions above and the Corollary for the number of Oracle calls in the final draft.
> >
> > Looks like I cannot edit my score right now, but I will champion this paper for acceptance.

---

> > > ### Author Response · Authors · 2024-11-25
> > > **Reply**
> > >
> > > We thank the reviewer for their reply and are glad they will champion the paper for acceptance. We will certainly include the discussions along with the Corollary in the revised draft.

---

### Official Review · Reviewer_m261 · 2024-11-08
**Review of Convergence in Phi-Divergence**

**Rating:** 5
**Confidence:** 4

**Review:**

*** Summary ***

The paper shows convergence results for both the unadjusted Langevin and proximal samplers in arbitrary $\Phi$-divergences, under a more general class of assumptions (the $\Phi$-Sobolev inequalities).

*** Strengths/Weaknesses ***

The work shows a convincing connection between $\Phi$-divergence and a class of $\Phi$-entropy inequalities.

The analysis nicely generalizes what is known for the Langevin dynamics/proximal sampler under more “standard” functional inequalities.

In my opinion, the paper is somewhat lacking motivation. Why might the reader be interested in Phi-divergences beyond those considered in e.g., Vempala and Wibisono or Chewi et al. 2022? The R\’enyi and KL divergences already imply results for many distributions of interest. I hope the authors can point to a few cases where we can hope to either get by with weaker results under more general assumptions, or stronger results with more restrictive assumptions. I hope the authors can emphasize these aspects in a revision.

If instead the authors want to focus on their proof techniques, it would be interesting if they could extend it to broader classes of Markov chains. See my question below.

Secondly, I think the proof technique can largely be found in prior work. For instance, Lemma 14 and 15 result immediately from an application of the Phi-Sobolev inequality to the decay equations, which can be found in Chewi (2024).

*** Minor/Questions***

Can one demonstrate similar results for chains after metropolisation? What about for the underdamped dynamics?

How can the results of [1] be placed in this context? Alternatively, what about the LOI results in Chewi et al 2022? (There is some discussion about the relationship between Phi-entropy and LOI in Bakry, Gentil and Ledoux.

I am not aware of Dalalyan 2017 having results in KL divergence. Perhaps the authors just mean in W2. Secondly, Chewi et al. 2022 (a) does not contain any mixing results to $\nu$, but only $\nu^\eta$.

Pg. 2 Assume smoothness assumptions -> make smoothness assumptions

Pg. 4 based on Metropolis … -> based on the Metropolis …

Pg. 4 from stationary distribution -> from the stationary distribution

The final equation on Page 7 has a sign error in the first equality.

Page 9: obtained in -> obtained is

[1] Mousavi-Hosseini, Alireza, et al. "Towards a complete analysis of langevin monte carlo: Beyond poincaré inequality." The Thirty Sixth Annual Conference on Learning Theory. PMLR, 2023.

**Paper Award:**

No

---

> ### Author Response · Authors · 2024-11-23
>
> We thank the reviewer for their feedback and careful reading of the manuscript. We would like to point to our rebuttal for Reviewer W2UN, in particular the first two points. We discuss comparisons between $\Phi$-Sobolev inequalities and mention some corresponding implications of our results to mixing time. We address the other comments below.
>
>
> * As the reviewer points out, many mixing implications have been shown in specific results in previous works, for example in [Vempala, Wibisono 23] or [Chen, Chewi, Salim, Wibisono 22]. One of our contributions is to provide a unified analysis technique via the information-theoretic SDPI approach that directly generalizes previous mixing results to all $\Phi$-divergences for strongly log-concave targets. We believe that highlighting the common structure in many previous proofs is beneficial to conceptually simplify the various sampling algorithms and mixing results.
>
> * We would like to mention that our SDPI-based proof technique and results also hold for a broad class of Markov chains defined by a pushforward and convolution step, e.g., where in each step $\rho_{k+1} = F$#$\rho_k * \mathcal{N}(0, \Sigma)$, where $F$ is a deterministic map which is a contraction ($\|F\|_{\textsf{Lip}}\leq 1$), and $\Sigma$ is a PSD matrix. The ULA falls in this class, and so does the underdamped (kinetic) Langevin algorithm, after an appropriate change of coordinates. Therefore, our results do go through for the underdamped Langevin case. In this paper, we decided to focus on the overdamped Langevin dynamics and its discretizations as they form a natural, popular, and simple set of Markov chains to study. We plan to investigate these generalizations in more detail in a future work. However, if the reviewers think it is crucial, we can also include this generalization in a revision of this paper.\
> Currently, our technique does not handle a Metropolis adjustment step, and it would be interesting to study how to extend our results to that case.
>
> * The usual Poincaré inequality implies the weak Poincaré inequality setting studied in [Mousavi-Hosseini et al. 23] and the other direction does not hold. As Section 2.2 in [Chafaï 04] shows that the Poincaré inequality is the weakest $\Phi$-Sobolev inequality, in that it is implied by any other $\Phi$-Sobolev inequality, our results do not include the regime studied in [Mousavi-Hosseini et al. 23]. A brief discussion on Section 2.2 from [Chafaï 04] is also included in the first bullet in our rebuttal to Reviewer W2UN.
>
> * To the best of our knowledge, [Chewi, Erdogdu et al. 22] study mixing to the target distribution $\nu$ under it satisfying an LOI inequality so it is not directly comparable to our results for ULA as we study mixing to $\nu^{\eta}$. As far as the implications between the functional inequalities are concerned, LOI interpolate between log-Sobolev inequalities (LSI) and Poincaré inequalities (PI), but $\Phi$-Sobolev inequalities go beyond that regime. Within the ''between LSI and PI'' regime, Beckner inequalities [Section 7.6.2 in Bakry, Gentil, Ledoux 14] interpolate between LSI and PI and are an instance of $\Phi$-Sobolev inequalities. LOI and $\Phi$-Sobolev inequalities however seem to be incomparable, as LOI do not correspond to a specific family of $\Phi$.
>
> * We thank the reviewer for pointing out that [Dalalyan 17] only covers results in Wasserstein distance; we will fix this in our revision.
>
> * We thank the reviewer for their careful reading of our paper; we will fix the typos in our revision.
>
>
> [Vempala, Wibisono 23] Santosh S. Vempala and Andre Wibisono. Rapid Convergence of the Unadjusted Langevin Algorithm: Isoperimetry Suffices. Geometric Aspects of Functional Analysis, 2023.
>
> [Chen, Chewi, Salim, Wibisono 22] Yongxin Chen, Sinho Chewi, Adil Salim, and Andre Wibisono. Improved analysis for a proximal algorithm for sampling. In Conference on Learning Theory, pages 2984–3014. PMLR, 2022
>
> [Mousavi-Hosseini et al. 23] Alireza Mousavi-Hosseini, Tyler Farghly, Ye He, Krishnakumar Balasubramanian, Murat A. Erdogdu. Towards a complete analysis of langevin monte carlo: Beyond poincaré inequality. The Thirty Sixth Annual Conference on Learning Theory. PMLR, 2023.
>
> [Chafaï 04] Djalil Chafaï. Entropies, convexity, and functional inequalities, On $\Phi$-entropies and $\Phi$-Sobolev inequalities. In J. Math. Kyoto Univ. 44(2): 325-363 (2004).
>
> [Bobkov, Gentil, Ledoux 14]  Bakry, Dominique, Ivan Gentil, and Michel Ledoux. Analysis and geometry of Markov diffusion operators. Vol. 103. Cham: Springer, 2014.
>
> [Chewi, Erdogdu et al. 22] Sinho Chewi, Murat Erdogdu, Mufan Li, Ruoqi Shen, and Matthew Zhang. Analysis of langevin monte carlo from poincare to log-sobolev. Foundations of Computational Mathematics (2024): 1-51.
>
> [Dalalyan 17] Arnak Dalalyan. Further and stronger analogy between sampling and optimization:
> Langevin Monte Carlo and gradient descent. In Conference on Learning Theory, pages
> 678–689. PMLR, 2017

---

### Official Review · Reviewer_W2UN · 2024-11-09
**Clear message and contributions**

**Rating:** 7
**Confidence:** 4

**Review:**

The paper provides convergence bounds in $\Phi$-divergence for two discrete-time Markov chain algorithms, the unadjusted Langevin algorithm (ULA) and the proximal sampler, for the class of all twice-differentiable strongly convex $\Phi$. The main results are presented as Theorem 1 and Theorem 2, and the bulk of the work focuses on explaining and outlining the proofs. The key technical tools are
- Definition 3, which define the Phi-Sobolev inequality used as assumptions to the main results,
- Lemma 7, which relates the rate of change of Phi-divergence along a SDE regardless of its drift, and
- Definition 10, which defines a strong data processing inequality in terms of Phi-divergence and whose contraction term is controlled throughout the subsequent proofs.

Strengths:

- I find the paper very well-written. It not only provides a clear contribution but also does a very good job at providing the relevant literature context and explaining the complex technical steps

- While $\Phi$-divergence results for continuous-time ULA, as well as special cases of $\Phi$-divergence results for both ULA and proximal samplers, are known, general $\Phi$-divergence resutls for discrete-time settings are not available to the best of my knowledge, and I think this paper provides concrete contributions in this direction. I also find the authors' unifying view of known results and proof techniques through Phi-divergence very satisfying; the exposition has done a very good job in relating results that use log-Sobolev inequality / Poincare inequality and that use different metrics together, and allows readers to understand its shared mathematical structure.

Minor comments:

- The main results seem to convey that, depending on the Phi-sobolev inequality satisfied by the stationary distribution, the algorithms will enjoy convergence guarantees with different strengths (in the sense of convergence in different d_Phi's). While it is easy to see how the strengths of d_Phi's compare according to different Phi's (e.g. KL is dominated by chi-squared), comparing the strengths of different Phi-sobolev inequalities is a bit less obvious. I would appreciate it if the authors can say a word or two about this, which would allow for a better appreciation of the general results beyond the common settings of KL and chi-squared divergences

- I also feel that the paper's contributions would have been stronger, if the authors can show that this unifying view through Phi-divergences allows for new discrete-time results through other metrics that were previously unknown. However I do understand that this can be difficult.

- In the second paragraph below Theorem 1, the authors say "It is interesting to ask for .... weaker assumptions on $\nu$ (See Section 5)... Therefore one can ask if ... when $\nu$ undergoes a suitable perturbation." The wording is slightly confusing: It seems to suggest that additional results on this are available in Section 5, but Section 5 is just a discussion section, and if I understand correctly, neither the paper nor the appendix actually provides such results. I think it'd be great if the authors can make it clearer that this is a point of discussion / future work rather than something in the paper.

- I wonder whether there's an intuitive explanation of why the drift term b_t disappears in Lemma 7. As commented by the authors at the bottom of page 11, this plays a role into the results later on. While I appreciate that the algebraic terms cancels by inspecting the proof, I wonder whether there's some intuition which would help readers appreciate Lemma 7 better, especially since it's a core tool of the paper. Again I understand that this may be difficult and not obviously doable.



All comments above are minor points that I think could improve the paper even further. Even if none of the above are addressed, I think the paper is clearly above the acceptance threshold and I am happy to see it being accepted into ALT.

**Paper Award:**

No

---

> ### Author Response · Authors · 2024-11-23
>
> We thank the reviewer for their review and comments, and we are glad they found the paper well-written. We address their comments below.
>
> * We thank the reviewer for bringing up the point regarding comparing the strength of various $\Phi$-Sobolev inequalities. We will add a brief discussion on the relations between $\Phi$-Sobolev inequalities in the revised draft.\
> From Section 2.2 in [Chafaï 04], we can see that the Poincaré inequality is the weakest $\Phi$-Sobolev inequality, in that it is implied by any other $\Phi$-Sobolev inequality. This extends the more well-known statement that the log-Sobolev inequality implies the Poincaré inequality. Furthermore, the same section also states that one can start from a $\Phi$-Sobolev inequality and show that it satisfies a $\Phi$-Sobolev inequality for $\Phi(x) = x^p -1$ where $p \in (1,2]$ but without a tight constant. This class of $\Phi$-Sobolev inequalities are related to Beckner inequalities [Section 7.6.2 in Bobkov, Gentil, Ledoux 14].
>
> * We would like to highlight that for strongly log-concave target distribution, our results directly imply fast mixing of both the ULA and the proximal sampler to the stationary distribution in all $\Phi$-divergences.\
> This is because an $\alpha$-strongly log-concave distribution satisfies all $\Phi$-Sobolev inequalities with the same optimal constant ($\alpha$).\
> In particular, this implies mixing in $\Phi$-divergences corresponding to $\Phi(x) = x^p-1$ for $p \in (1,2]$ (the corresponding $\Phi$-Sobolev inequality is the Beckner inequality [Section 7.6.2 in Bobkov, Gentil, Ledoux 14]).\
> Furthermore, we can also use the implications between $\Phi$-Sobolev inequalities mentioned in the bullet above to obtain interesting guarantees for the proximal sampler. In particular, as the log-Sobolev inequality implies the Beckner functional inequality, our results show mixing in ''Beckner divergence'' (i.e. $\mathsf{D}_{\Phi}$ for $\Phi(x) = x^p-1$ for $p \in (1,2]$) under a log-Sobolev inequality assumption on the target or stationary distribution for the proximal sampler.\
> We will add a discussion on these implications in the final version.
>
> * We thank the reviewer for their comment. Their understanding is correct and we will rephrase the text below Theorem 1 and Section 5.
>
> * One intuition we have is that the contraction of $\Phi$-divergence between two simultaneous Fokker-Planck evolutions is entirely due to the Brownian motion component.\
>     For example, from the exposition in [Section 8.3 in Chewi 24], we can see that the time-derivative of the simultaneous $\Phi$-divergence, i.e. the $\Phi$-Fisher information $\mathsf{FI}_{\Phi}$, can be written in terms of the carré du champ operator associated with the diffusion process, which is only determined by the Brownian motion component.\
>     We will mention these interpretations in the revision.
>
> [Chafaï 04] Djalil Chafaï. Entropies, convexity, and functional inequalities, On $\Phi$-entropies and $\Phi$-Sobolev inequalities. In J. Math. Kyoto Univ. 44(2): 325-363 (2004).
>
> [Bobkov, Gentil, Ledoux 14]  Bakry, Dominique, Ivan Gentil, and Michel Ledoux. Analysis and geometry of Markov diffusion operators. Vol. 103. Cham: Springer, 2014.
>
> [Chewi 24] Sinho Chewi. Log-concave sampling. 2024.

---

> ### Comment · Reviewer_W2UN · 2024-11-27
>
> I have read all other reviews and the authors' response, and I am happy to keep my rating for acceptance.

---

### Author Rebuttal · Authors · 2024-11-23

We have added rebuttals to each of the reviewers separately, as **Official Comments** to the reviews.

---

### Meta-Review · Area_Chair_zc4i · 2024-12-10

**Recommendation:** Accept
**Confidence:** 4

**Metareview:**

This work provides convergence of two discrete algorithms, unadjusted Langevin and the proximal sampler, for sampling distributions satisfying the $\Phi$-Sobolev inequality; where convergence is stated in the corresponding $\Phi$-divergence. In this sense, the submission extends and unifies multiple existing results in the literature, and provides a clean and insightful perspective on how to approach these questions.

The reviewers were mostly positive, and highly appreciated the clarity and quality of writing. Some criticism, related to the lack of fundamentally new techniques, was mentioned, but overall the positive aspects overcome the criticism. The rebuttal period was quite productive, and the authors agreed on including several clarifications and side results which I believe can help greatly with the value of this work.

**Paper Award:**

No